# Accessing habitat suitability and connectivity for the westernmost population of Asian black bear (*Ursus thibetanus gedrosianus*, Blanford, 1877) based on climate changes scenarios in Iran

**Maryam Morovati**[1,2]*, **Peyman Karami**[3], **Fatemeh Bahadori Amjas**[4]

**1** Department of Environmental Sciences & Engineering, Faculty of Agriculture & Natural Resources, Ardakan University, Ardakan, Iran, **2** Medicinal and Industrial Plants Research Institute, Ardakan University, Ardakan, Iran, **3** Department of Environmental Sciences, Faculty of Natural Resources and Environment Sciences, Malayer University, Malayer, Iran, **4** Faculty of Agriculture and Natural Resources, Ardakan University, Ardakan, Iran

* mymorovati@ardakan.ac.ir

## Abstract

Climate change, as an emerging phenomenon, has led to changes in the distribution, movement, and even risk of extinction of various wildlife species and this has raised concerns among conservation biologists. Different species have two options in the face of climate change, either to adopt or follow their climatic niche to new places through the connectivity of habitats. The modeling of interpatch landscape communications can serve as an effective decision support tool for wildlife managers. This study was conducted to assess the effects of climate change on the distribution and habitat connectivity of the endangered subspecies of Asian black bear (*Ursus thibetanus gedrosianus*) in the southern and southeastern Iran. The presence points of the species were collected in Provinces of Kerman, Hormozgan, and Sistan-Baluchestan. Habitat modeling was done by the Generalized Linear Model, and 3 machine learning models including Maximum Entropy, Back Propagation based artificial Neural Network, and Support Vector Machine. In order to achieve the ensemble model, the results of the mentioned models were merged based on the method of "accuracy rate as weight" derived from their validation. To construct pseudo-absence points for the use in the mentioned models, the Ensemble model of presence-only models was used. The modeling was performed using 15 habitat variables related to climatic, vegetation, topographic, and anthropogenic parameters. The three general circulation models of BCC-CSM1, CCSM4, and MRI-CGCM3 were selected under the two scenarios of RCP2.6 and RCP8.5 by 2070. To investigate the effect of climate change on the habitat connections, the protected areas of 3 provinces were considered as focal nodes and the connections between them were established based on electrical circuit theory and Pairwise method. The true skill statistic was employed to convert the continuous suitability layers to binary suitable/unsuitable range maps to assess the effectiveness of the protected areas in the coverage of suitable habitats for the species. Due to the high power of the stochastic forest model in determining

**Data Availability Statement:** All relevant data are within the paper and its Supporting Information files.

**Funding:** The authors received no specific funding for this work.

**Competing interests:** The authors have declared that no competing interests exist.

the importance of variables, this method was used. The results showed that presence/absence models were successful in the implementation and well distinguished the points of presence and pseudo-absence from each other. Based on the random forests model, the variables of Precipitation of Driest Quarter, Precipitation of Coldest Quarter, and Temperature Annual Range have the greatest impact on the habitat suitability. Comparing the modeling findings to the realities of the species distribution range indicated that the suitable habitats are located in areas with high humidity and rainfall, which are mostly in the northern areas of Bandar Abbas, south of Kerman, and west and south of Sistan-Baluchestan. The area of suitable habitats, in the MRI-CGCM3 (189731 $Km^2$) and CCSM4 (179007 $Km^2$) models under the RCP2.6 scenario, is larger than the current distribution (174001 $Km^2$). However, in terms of the performance of protected areas, the optimal coverage of the species by the boundary of the protected areas, under each of the RCP2.6 and RCP8.5 scenarios, is less than the present time. According to the electric circuit theory, connecting the populations in the protected areas of Sistan-Baluchestan province to those in the northern Hormozgan and the southern Kerman would be based on the crossing through the heights of Sistan-Baluchestan and Hormozgan provinces and the plains between these heights would be the movement pinch points under the current and future scenarios. Populations in the protected areas of Kerman have higher quality patch connections than that of the other two provinces. The areas such as Sang-e_Mes, Kouh_Shir, Zaryab, and Bahr_Aseman in Kerman Province and Kouhbaz and Geno in Hormozgan Province can provide suitable habitats for the species in the distribution models. The findings revealed that the conservation of the heights along with the caves inside them could be a protective priority to counteract the effects of climate change on the species.

## 1. Introduction

There is currently undeniable evidence that climate change affects the distribution, behavior, and biodiversity of plant and animal species [1, 2]. Climate change can lead to a change in the expansion of suitable habitats beyond the boundaries of protected areas [3]. Climate change, by changing the habitat of many species, leads to changes in their dispersal patterns, which can increase the level of conflict between humans and wild animals[4]. According to studies on 37 plant and animal species in Iran, more than 30 species will lose their suitable habitats due to climate change, which shows the importance of this phenomenon on the distribution of flora and fauna in the country [2]. As a country located mostly in an arid region, Iran has severely been affected by global climate change. According to the scientific reports, Iran will experience an average temperature increase of between 3.4–3.5° C by 2100 [3]. In response to climate change, there are two ways for a species either to adapt to the change or to pursue its desirable climatic conditions. Due to the acceleration of climate change, the choice of the second way seems more likely for many species [5]. Species movement among habitat patches is one of the best ways to reduce the effects of climate change [6]. The negative impact of climate change on large mammal species is very important because as they are more exposed to the surrounding climatic conditions than small mammals due to their large size and at the same time, their poor ability to use micro-climatic refuges (e.g. underground areas and vegetation cover) [7, 8].

Therefore, they are increasingly regulating their spatio-thermal distribution in response to the thermal constraints caused by rising temperatures [9]. The dispersal ability and habitat

connectivity are two very important factors in studying species movement pathways in search of a suitable climate [2]. Perhaps the reliable way to identify the movements of animal species is through field observations, but it is difficult or even impossible to collect such data [10]. Accordingly, a combination of field observations and simulation models seems to be more logical [11, 12]. Many methods have been used so far to identify the corridors of various species [11, 13–21]. The most important of which are Least-Cost Path (LCP) analysis, electric circuit theory [19], and centrality analysis [15].

Models with the ability to predict the suitability of wildlife habitats on a large scale can be useful for wildlife managers [22, 23]. For the protection of an important species, it is of critical importance to identify its needs as well as habitat constraints, and degradation factors [24, 25]. Such models by identifying these factors can help managers spend less time and money in the management of wildlife habitats [26]. Species distribution modeling is a set of approaches, definitions, and techniques, based on ecological and biogeographical concepts and describes the relationships between species distribution and their physical environment. These are quantitative and experimental models of species-environment relationships that are constructed using the data related to the species location (such as abundance and presence) and environmental variables that are assumed to affect the species distribution [27]. In addition to explaining the ecological conditions required for species and issues related to the species niche, these models produce distribution prediction maps, which are usually used as inputs for the other analyses, especially in protection planning projects [28, 29]. Large carnivores such as bears, due to their widespread distribution and low population size as well as human pressure are particularly sensitive to habitat fragmentation and loss [30].

Pressures including increasing population growth and density, conversion of habitats and human settlement, road construction [31], urban planning and land use change [32] hunting [33], entry of non-indigenous species [34], climate change [3], and deforestation [35] are some of the existing concerns threatening the species biodiversity.

Existence of biodiversity depends on having sustainable environmental factors [36]. Today, with the increase of human manipulation in natural resources and the environment, the sustainability of environmental factors is under threat for various reasons, especially climate change [37]. Protected areas are now considered the last refuge for wildlife species. The negative impact of climate change on the borders of these areas and the optimal coverage of the species habitats has been one of the important fields of recent studies [38–40].

Estimating "how much" and "where" populations are most at risk is paramount to understand the effects of climate change on different species. It is also critical to assess whether these populations are safe in different parts of the species range. That is why it is required to pay attention to protected areas [41].

Climate change may produce species' range shifts altering patterns of species' diversity and distribution, that could significantly reduce conservation efficiency of networks of protected areas [42]. Therefore, it is necessary to revise the boundaries of protected areas using model species such as wild sheep [3].

The Asian black bear (*Ursus thibetanus*, G. Cuvier, 1823) is one of the largest carnivores on Earth [43]. It is a medium-sized bear and includes seven subspecies in the world [44]. Populations of this species have declined by 30 to 40% over the past 30 years [45]. The subspecies of Asian black bear (*Ursus thibetanus gedrosianus*, Blanford, 1877) reaches the westernmost distribution limit of this species. The subspecies is distributed in Iran and Pakistan and is also known as Balochistan black bear. The distribution of this species is limited to the mountainous areas around Sistan-Baluchestan, Hormozgan, and Kerman Provinces [46]. Although the Asian black bear is classified as a vulnerable species based on IUCN red list [45], the subspecies

of Iranian black bear is highly endangered due to the population decline and habitat isolation [47, 48].

Due to local extinctions of this species in some areas of its distribution [49] and habitat destruction [50], the fragmentation and isolation of habitats of this species has been approved in the south of Iran. In addition to the mentioned factors, the proven impact of climate change on Iran's wildlife, on one hand [2] and the impact of climate variables on its distribution range on the other hand [49] are the main reasons the climate change should be considered as a concern for this species.

The habitats of this species are the mountainous and forest areas in the south of Kerman Province, and the dispersed habitats of *nannorrhops ritchieana* shrubs and olive trees in Sistan-Baluchestan and Hormozgan Provinces. They come out of their den at late dusk and get back before sunrise [46]. They mostly feed on herbs, such as fruits and sprouts [51].

Black bears have a good capacity of adaptation to different environments and foods; however, due to the accessibility of the animal's hideout, hunters can easily find and kill them [52].

Currently, black bears, inhabiting in the forest areas of Jebalbarez and Delfard, are on the verge of extinction, and a partial decrease in their population has been witnessed in other areas, as well. Numerous studies have been performed on the Asian black bear (*Ursus thibetanus*) [53–56], but the studies on the subspecies of the black bear in Iran (*Ursus thibetanus gedrosianus*) are very limited in number and single-focused [49, 57]. So far, no comprehensive study has been conducted on the effect of climate change on the habitat connectivity and efficiency of the protected areas for these species under different climate change scenarios.

Considering the issues raised, this study seeks to investigate the effect of climatic parameters on the distribution of black bears under the different climate change scenarios, identify their communication paths, and finally investigate the efficiency of protected areas to include the species' preferred habitat in southern Iran.

## 2. Materials and methods

### 2.1. Study area and species occurrences

The present study was conducted in Kerman, Sistan-Baluchestan, and Hormozgan Provinces in the southern Iran, as the main habitats of the black bear in the country (Fig 1). Additional information on the areas is provided in Table 1. First, by reviewing the studies conducted on this species [46, 58] and the reports from the Department of Environment (DoE) of Kerman, Sistan-Baluchestan, and Hormozgan provinces, the geographic range of the species was determined. Field surveys (from early April 2018 to early April 2019) were then conducted to record the presence points. During the field visits, all observations related to footprints, feces, resting place, and/or signs remaining on trees were considered as the species presence points. In total, 61 presence points were registered for the species in Kerman Province. In Hormozgan Province, in addition to covering the eastern parts of Bashagard and Roudan Counties, the species were also surveyed in the northern parts, including Hamag Mountain, Shagheh Roud, and Geno Protected Area, and only 13 points were collected in total. In Sistan-Baluchestan Province, a wider range was chosen for the survey, as black bears are more widespread in the province, according to the reports from the DoE. In most areas with date palm vegetation, the species is more likely to be present [59]. Therefore, all areas of Nikshahr (heights of Shalmal, Abband and Darreh Zendani, Kooshat, Jowzdar and Malouran, Karchan, Kuh-e Naran, and Kahiri), Sarbaz, Kuh-e-Birk, Saravan, Khash, and Bazman of Iranshahr were surveyed and a total number of 21 presence points were found (Fig 1).

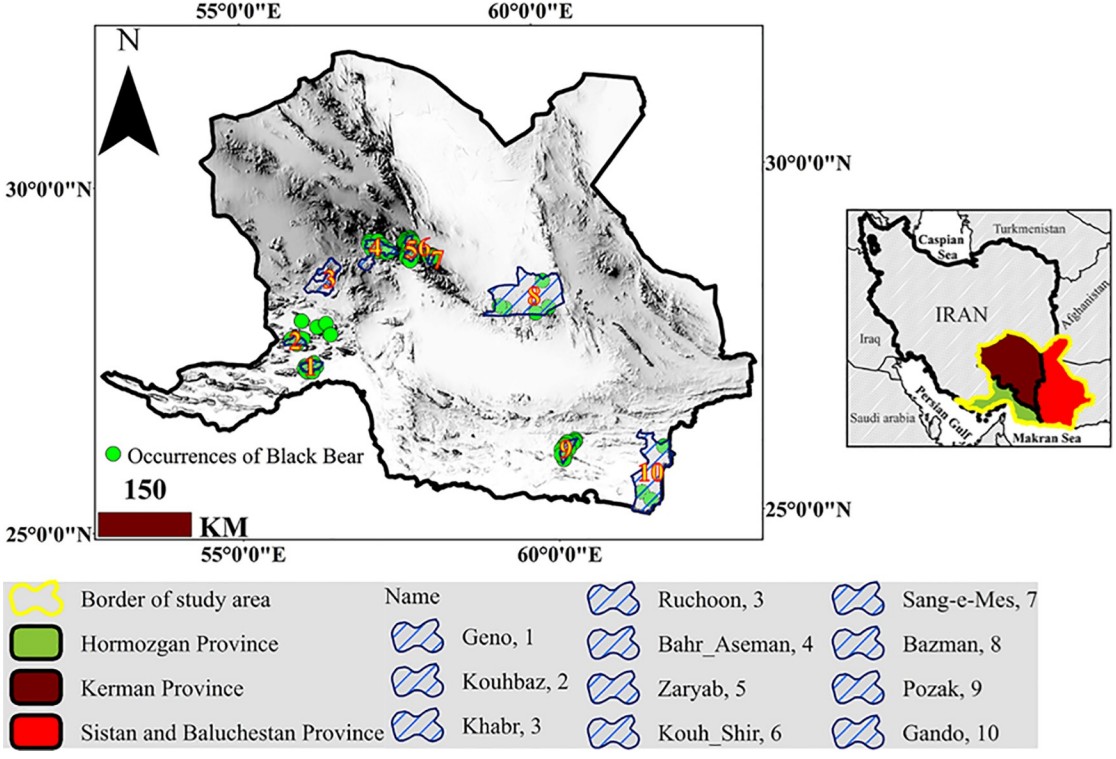

**Fig 1. Location of the study area, presence points, and protected areas.**

## 2.2. Environmental variables

By reviewing the studies performed on this species [46, 49, 53, 57] climatic, topographic, anthropogenic, and vegetation variables including elevation, distance from gardens, main roads, streams (waterways), human settlements, landscape fragmentation index, cluster hill shade of elevation, and average Normalized Difference Vegetation Index (NDVI) were used as the model inputs. The elevation variable was prepared with an approximate accuracy of one kilometer from the WORLDCLIM database [60]. Since, according to previous studies, gardens would be a stimulus for the species presence [46], the distance from gardens was selected as an input variable and prepared from the land use/land cover map of the study area, which had already been prepared and made available by Forests, Rangeland and Watershed Management Organization of Iran (FRWMO). The cluster hill shade of elevation was another criterion for analyzing solar energy received by the bears in their habitats and niche [61]. The hill shade values represent the average amount of shadow that each cell of the region receives over the year. Areas that receive less sunlight during the day will have more moisture. The hill shade map was prepared from Digital Elevation Map (DEM) using the Terrain Mapping toolbox in the ArcGIS 10.4.1 [62, 63]. The NDVI was used to examine the vegetation cover of the study area. The map of this variable was prepared from MODIS / 006 / MYD13A1 product of Aqua Modis sensor with 500 m spatial resolution and 16-day repetition frequency in Google Earth Engine System [64, 65]. Using the average method, the average of annual NDVI data was obtained to eliminate the effect of healthy and abnormal weather on the status of vegetation cover [64]. The landscape fragmentation index was also obtained from land use/cover map by applying a 3×3 filter on it in the Idrisi TerrSet software. The index fluctuates in the range of 0 (no fragmentation) to 1 (complete fragmentation) [66].

**Table 1. Characteristics of the protected areas (focal nodes) in kerman, hormozgan and Sistan-Baluchestan Provinces.**

| No. | Province | Areas studied | Geolocation of the study areas |
|---|---|---|---|
| 1 | Kerman | Bahr Aseman Protected Area | Bahr Aseman Protected Area covers 118800 hectares, and is confined to 2846-2913N to 5656-5735E. It is located 215 kilometers south of the center of Kerman Province, and 25 kilometers east of the city Rabor, and 40 kilometers north west of the city Jiroft. The black bear is one of the animal assets of the region with its presence in the central and northern parts. |
| 2 | | Sang-e Mes Protected Area | Sang-e Mes Protected Area, with an area of 10316 hectares, is confined to 2850-2858N to 5803–811 E. It is located in the east of Kerman Province, 205 kilometers southeast of the province capital city, and 30 kilometers south of the city Bam. The most prominent animal species in the protected area is the Asian black bear. In recent years, some of the natives have occasionally confirmed the presence of the bears in the northern parts. |
| 3 | | Kouh Shir Protected Area | Kouh Shir Protected Area, with an area of 58519 hectares, is confined to 2859- 2809N to 5754-5805E. It is located 170 kilometers southeast of the capital city of the province. |
| 4 | | Zaryab Wildlife Refuge | Zaryab Wildlife Refuge covers an area of 45250 hectares and is confined within 2856-2915N to 5749- 5754E. It is located 150 kilometers southeast of the capital city of Kerman Province. One of the most important species in Zaryab Wildlife Refuge is Asian black bear. |
| 5 | | Ruchoon Wildlife Refuge and Khabr National Park | Khabr National Park and Ruchoon Wildlife Refuge are confined to 2859-2825N and 5602-5639E. Khabr National Park covers an area of 149982 hectares and Ruchoon Wildlife Refuge covers an area of 28171 hectares. Asian black bears in these regions are limited in number and disperse over a small area; however, detailed information on their number is not available. |
| 6 | Sistan-Baluchestan | Pozak Protected Area | Pozak Protected Area, with an area of 46144 hectares, is located in 275550N to 60013E, in Sistan-Baluchestan Province. The main habitat of the black bear exists in this province. |
| 7 | | Bazman Protected Area | Bazman Protected Area is located in 6000E to 2804N, in Sistan-Baluchestan Province. It covers an area of 324688 hectares. Black bears have occasionally been observed in this area. |
| 8 | | Gando Protected Area | Gando Protected Area is located in 6125E to 2605N in Sistan-Baluchestan Province. It is located along Iran-Pakistan border to the southeast. Black bears are rarely found in this area. |
| 9 | Hormozgan | Geno Biosphere Reserve | Geno Biosphere Reserve is located in 2724N to 5607E, in Hormozgan Province. This protected area is 29 kilometers northwest of the city Bandar Abbas. |
| 10 | | Kouhbaz Protected Area | Kouhbaz Protected Area is in 5535-5601E to 2743-2748N in Hormozgan Province. It covers an area of 34109 hectares. |

In addition to the mentioned habitat variables, the current climatic variables (time period: 1979–2013) were downloaded from CHELSA database with high accuracy (30 arcsec, CH 1 km) [67]. Three General Circulation Models (GCMs), including CCSM4 (National Center for Atmospheric Research, USA), BCC-CSM1-1(Beijing Climate Center, China Meteorological Administration), and MRI-CGCM3(Meteorological Research Institute, Japan) have been proved to be suitable for the modeling of climate change in Iran and provide a better forecast of species dispersal than the other GCMs [68–71]. Therefore, these three models were used to

**Table 2. Habitat variables used to assess the effect of climate change on habitat suitability and movement of black bears in southern Iran.**

| No. | Variable | Range | Period | Unit |
|---|---|---|---|---|
| 1 | Altitude | -4-4310 | Current | Meters |
| 2 | Distance from garden | 0–198776 | Current | Meters |
| 3 | Distance from main road | 0–102359 | Current | Meters |
| 4 | Distance from streams | 0–35746 | Current | Meters |
| 5 | Distance from Human Statement | 0–122620 | Current | Meters |
| 6 | Landscape fragmentation | 0–0.75 | Current | - |
| 7 | Cluster Hill shade of Elevation | 185–255 | Current | - |
| 8 | NDVI | -0.12–0.60 | Current | - |
| 9 | Isothermally(Bio3) | 272–417 | Current-RCP2.6-RCP8.5 | Dimensionless |
| 10 | Temperature Annual Range(Bio7) | 196–408 | Current-Rcp2.6-Rcp8.5 | Degrees Celsius |
| 11 | Mean Temperature of Coldest Quarter (Bio11) | -76–217 | Current-RCP2.6-RCP8.5 | Degrees Celsius |
| 12 | Precipitation Seasonality (Bio15) | 57–138 | Current-Rcp2.6-Rcp8.5 | Fraction |
| 13 | Precipitation of Driest Quarter(Bio17) | 0–23 | Current-RCP2.6-RCP8.5 | Millimeters |
| 14 | Precipitation of Warmest Quarter(Bio18) | 0–82 | Current-RCP2.6-RCP8.5 | Millimeters |
| 15 | Precipitation of Coldest Quarter(Bio19) | 20–272 | Current-RCP2.6-RCP8.5 | Millimeters |

assess the current dispersal status of the bears and identify future corridors. Under the two severe and minor scenarios, the Representative Concentration Pathway (RCP) of RCP 8.5and RCP 2.6 were selected. Prior to modeling, the correlation between the habitat variables was checked and those variables with a correlation value of higher than 0.75 were excluded from the analysis [72]. The correlation test was performed in Arcgis10.4.1 software and using the Band Collection Statistics command from the Spatial Analyst Tools. After analyzing the correlation between the environmental variables, 15 variables were entered the modeling process as presented in Table 2.

## 2.3. Habitat suitability modeling

In order to model the species distribution, GLM (Generalized Linear Model) [73] and 3 machine-learning methods including Maxent [74], SVM [75], and BP-ANN [76, 77] were used in ModEco software [78]. To model the species distribution, 80% of the data were considered for training, and 20% for the test.

The validation of the models was tested using the species prevalence-independent metrics of TSS [79] and Area under ROC (receiver operating characteristic) curve (AUC) [80, 81]. The AUC values of less than 0.7 are considered suitable, 0.7 to 0.9 good, and above, very well [82]. The TSS values less than 0.2 indicate poor performance, 0.2 to 0.6 relatively good performance, and more than 0.6, suitable performance [83].

According to the calculated AUC for each model, the H0 hypothesis was also assessed to evaluate the performance of AUC as a diagnostic test. Therefore, H0, equality of sensitivity rate and specificity at the AUC of 0.5 was considered as the random performance of the model.

The TSS [84] was used to convert the continuous maps of habitat suitability to the binary maps (suitable/unsuitable). After identifying the threshold, the metrics of sensitivity, specificity, correct classification, and miss classification were used to assess the power of the threshold [3, 85].

Calculations related to the mentioned criteria were performed in SPSS.v16 software. Since the individual models can have different results [86], the Ensemble modeling approach was

used to integrate these models. Ensemble not only decreases uncertainty [87] but also reduces the weakness of individual models [88, 89]. In order to build the Ensemble model, the "accuracy rate as weight" method was used based on the AUC values.

Habitat suitability models require bias-free samples (species locality data) to run. In order to reduce the sample bias, autocorrelation of the presence points was examined at the distances of 1, 3, and 5 km. For this purpose, the topographic heterogeneity was calculated from DEM, and then, the points with spatial autocorrelation were removed by SDM toolbox within the mentioned distances [90]. Considering the distance of 5 km, only 53 points out of 95 points remained and were entered the modeling (Fig 2).

The models used in this study belonged to the presence/absence models that need the absence points or pseudo-absence to perform. There are several ways to create pseudo-absence points, including 1) random selection, 2) selection within a delimited geographical interval from the recorded presence points, and 3) selection based upon environmental variables [91]. In the third method, the relationship between environmental variables and presence points was established by models such as ENFA, BIOCLIM, and Domain. By creating the intermediate models, these models identify areas suitable for recording pseudo-absence points. Since false pseudo- absence points can affect the outputs [92], to increase the accuracy of pseudo-absence records, the output of presence-only models was used [93]. This is more standard than the other methods of preparing pseudo-absence points [91]. Three Domain, BIOCLIM, and One-CLASS SVM models were used to create pseudo-absence points. Totally, 53 presence points were entered the ModEco software using cross-validation and uncorrelated environmental variables. The validation results showed that all the three models had an acceptable validity. According to the results, the AUC value for BioCLIM, Domain, and One-Class SVM models was 0.92, 0.96, and 0.80, respectively. The three models were integrated to form the

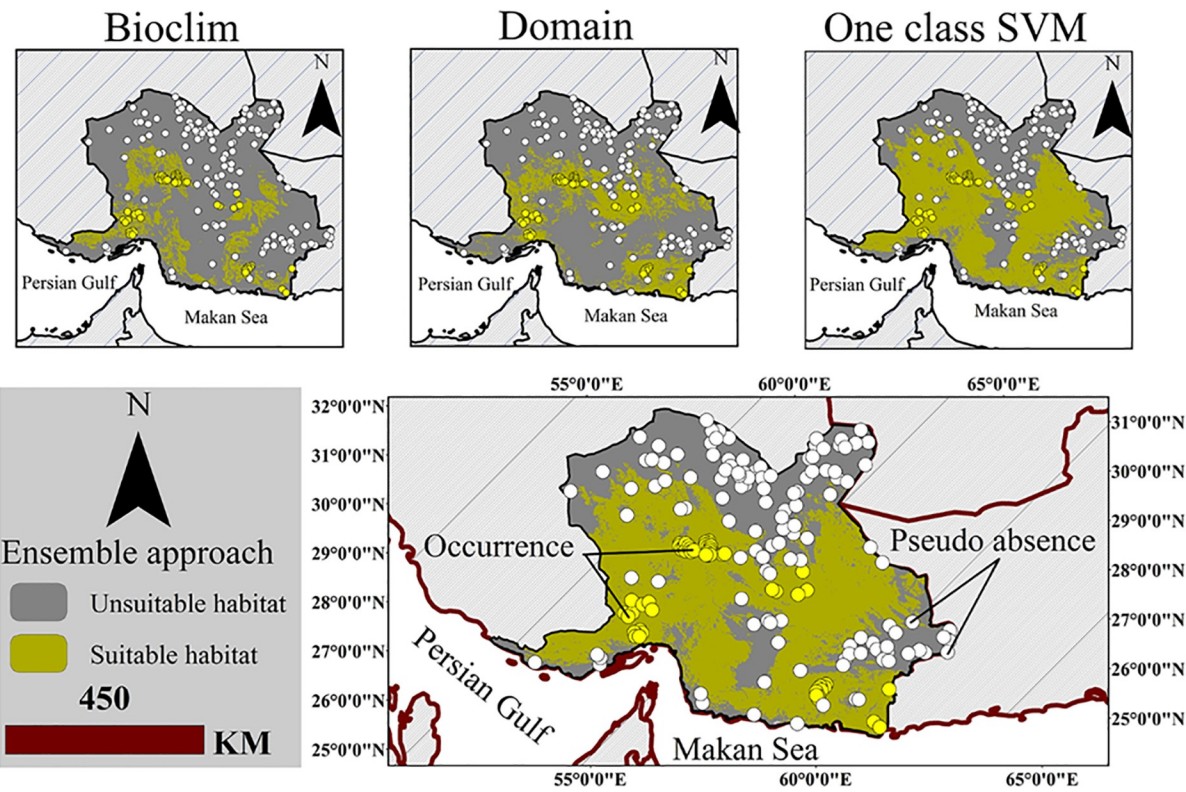

**Fig 2. Location of presence/pseudo-absence points in presence-only habitat suitability models.**

Ensemble binary approach. Then, pseudo-absence points were obtained by observing a minimum distance of 5 km at 2 times the presence points [94, 95] using the Hawats tools in the ArcGIS10.4.1 (Fig 2). Since the adaptation to the current and future climate change is one of the most important functions of conservation biology [96], The use of threshold is one of the methods of quantitative calculation of habitat suitability that is used to evaluate the efficiency of different boundaries [86, 97, 98]. It is possible to use different thresholds [99]. In this study, TSS was used as the threshold limit [3]. After identifying the cut-off value using TSS statistics, the Ensemble model were converted into binary maps and the area of suitable habitats in different scenarios was determined. Accordingly, the efficiency of the boundaries of protected areas in optimal coverage of suitable habitats in different climate change scenarios was assessed.

## 2.4. Landscape connectivity

In this study, the electric circuit theory [19] was used in the Circuitscape4.0 software (www.circuitscape.org) to identify the potential connection paths. This method provides a detailed exploration of potential linkage and connectivity variability [86]. Circuitscape software is able to translate spatial data set into a graph structure efficiently by converting cells to nodes and connecting them to their neighbors by resistors. This method produces maps that show the current flow at each cell in the landscape. A higher flow of the current between habitat patches emphasizes the spaces in which the species are more likely to move. More connectivity between habitat patches will be provided if multiple pathways are available. Places where current density is high or alternative routes are not available show pinch points that act as bottlenecks to movement. They reflect conservation priorities because losing them greatly affects the connectivity [19]. In order to implement this method, a cost map is required. Landscape resistance has frequently been used to bridge knowledge gaps on animal movements and is often the main core in the connectivity modeling associated with conservation initiatives.

It is worth noting that landscape resistance estimation is usually performed by parameterizing environmental variables across a 'landscape resistance' or 'cost to movement sequence". A low landscape resistance indicates the ease of movement, whilea high landscape resistance shows limited movements, or is used to denote a barrier towards movements. 'Friction' and 'impedance' to movement" or conversely, 'permeability' and 'conductivity' to movement are also terms used to describe travel surfaces [100, 101].

The cost or Landscape resistance map is displayed in raster format. Each cell on the map is assigned a numeric value representing the relative cost that must be paid to pass through each cell and can be calculated by combining different criteria [102]. To determine the cost of displacement in the landscape, the resistance map was prepared by inverting the map of habitat suitability [69, 86, 103]. With this operation, the lowest displacement costs will be assigned to those areas with the highest habitat suitability for the species, while the highest costs will be in habitats with low suitability [104]. The theory of electrical circuitry requires nodes to run. These nodes can be considered as the cores of suitable habitats identified by applying a threshold [49, 69, 86] or as the boundary of protected areas [105, 106].

In this study, protected areas were considered as focal nodes (Fig 1) and the probability of movement and displacement between all pairs of the protected areas was calculated by pairwise connectivity index and cumulative current flow [69]. Ruchoon and Khabar Protected Areas in Kerman Province were considered as one node due to their proximity to each other.

## 2.5. Importance of influential variables in the modeling

Modeling methods can be generally divided into two categories of parametric and nonparametric. Parametric methods have limitations such as "considering a default distribution

for the dependent variable", "linearity of the proposed relationship", "uniform variance of errors", and "independency of the variables" [107]. However, nonparametric methods do not have these limitations [108]. It should be noted that the analysis of complex ecological data requires robust and flexible analytical methods that control nonlinear relationships, interactions, and missing data. Therefore, in this study, non-parametric random forest classification method was used to determine the importance of habitat variables on the presence/absence of species. Random forest is a modern type of tree-based method that includes a multitude of classification and regression trees [109] that can be used for both classification and regression [110–113]. In this model, the importance of the variables entered into the model is measured based on two criteria, mean decrease in accuracy and mean decrease in Gini [114]. This feature of the model has led to its use in many studies to measure the importance of variables [115–119]. There are other advantages that have led to the increase in the usefulness of this model.

An important reason for their popularity is the availability of methods for determining the importance of variable [120]. They can handle large numbers of variables with relatively small numbers of observations and assess the importance of variables [110, 119, 121]. Random forest method was run in R3.5.2 software [110] to determine the effect of each variable on the suitability of the predicted areas in the habitat modeling [122].

## 3. Results

### 3.1. Model validation

Table 3 shows the validation results of the models. Based on the AUC values, the models were able to well distinguish between presence and absence points. Considering the P-values, the H0 hypothesis for the models mentioned in Table 3 was rejected at the significance level of 0.95. The models showed a significant difference in the AUC value with the random model (P-value <0.0001), which indicates the efficiency of the used models. The highest AUC value was found in the Ensembel model and the lowest in the MaxEnt model.

The TSS index falls within the acceptable range in all models, except for MaxEnt with a value of 0.82, reflecting the efficiency of the models. Therefore, the results of the modeling were confirmed by the criteria independent from the threshold.

### 3.2. Habitat suitability

Fig 3 shows a continuous map of either the species presence or the likelihood of suitability. Accordingly, blue colors indicate areas with high and desirable presence and brown colors indicate areas with low habitat suitability and low presence.

Among the different RCP 2.6 scenarios, the southeastern and western parts of Kerman Province have high suitability for the species, which corresponds to the Khabr National Park, Ruchoon Wildlife Refuge, Bahr_Aseman, Zaryab, Shir_Kooh, and Sang-e Mes.

**Table 3. Performance of the models used in the modeling process.**

| | | | Current Models | | | |
|---|---|---|---|---|---|---|
| Model | TSS | AUC | Standard error | Lower bound (95%) | Upper bound (95%) | p-value (Two-tailed) |
| MaxEnt | 0.82 | 0.957 | 0.013 | 0.932 | 0.982 | < 0.0001 |
| GLM | 0.90 | 0.979 | 0.006 | 0.968 | 0.990 | < 0.0001 |
| SVM | 0.90 | 0.986 | 0.003 | 0.980 | 0.992 | < 0.0001 |
| BP-ANN | 0.90 | 0.975 | 0.009 | 0.957 | 0.994 | < 0.0001 |
| Ensemble | 0.91 | 0.988 | 0.005 | 0.974 | 0.995 | < 0.0001 |

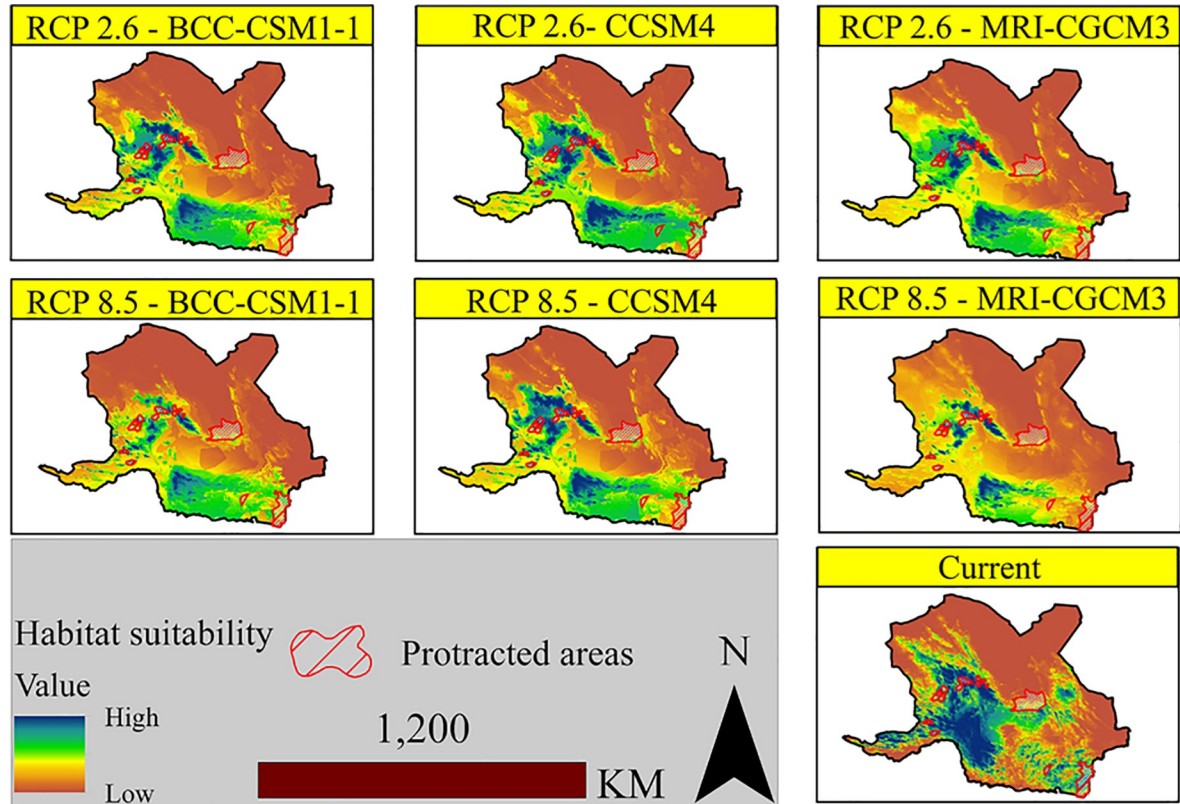

**Fig 3. Habitat suitability of Asian black bears under different scenarios based on the Ensemble modeling approach.**

According to the results, in MRI-CGCM3 model under the scenario RCP8.5, there is the greatest decrease in habitat suitability or probability of presence. In other words, in this scenario, the habitat suitability is more likely to decrease than the other two models of this scenario. In the same scenario, under CCSM4 model, there is a possibility of species presence in the north of Hormozgan Province and the south of Kerman Province. However, based on the results, there is a significant difference between the current distribution ranges of the predicted climate scenarios. The highest correlation between the high quality habitats of the current distribution ranges was found in this model and under this scenario.

The results from applying the threshold on the habitat suitability map are shown in Table 4. Based on the results, the threshold in most of the models was able to distinguish well the suitable and unsuitable habitats. Among the models used for the current state, MaxEnt had a lower Kappa index than the other models. Based on the results of threshold, this model was able to identify 0.92% of the presence points (sensitivity); however, the resolution of this

**Table 4. Threshold-dependent measures for the current and generalized models.**

| Model | threshold | Kappa | Sensitivity | Specificity | Correct classification | Misclassification |
|---|---|---|---|---|---|---|
| MaxEnt | >0.26 | 0.78 | 0.92 | 0.82 | 0.90 | 0.095 |
| GLM | >0.41 | 0.89 | 0.94 | 0.96 | 0.95 | 0.045 |
| SVM | >0.26 | 0.88 | 0.96 | 0.94 | 0.95 | 0.50 |
| BP-ANN | >0.04 | 0.87 | 0.98 | 0.92 | 0.94 | 0.056 |
| Ensemble | >0.28 | 0.90 | 0.98 | 0.93 | 0.95 | 0.050 |

model was not adequate enough to identify the absence points (specificity). The Ensemble modeling approach was able to identify the presence points with an accuracy of 0.98 and its accuracy for identifying the absence points was 0.93. The correct classification was also calculated to be 0.95.

After calculating the TSS, its value was applied to the continuous habitat suitability maps. Fig 4 shows the suitable/unsuitable habitats under the different climate change scenarios. The red and green colors on the maps represent suitable and unsuitable habitats, respectively.

Table 5 shows the area of suitable habitats inside and outside the protected areas under different scenarios. Under the RCP2.6 scenario, the area of suitable habitats is higher in the MRI-CGCM3-2.6 model compared to the other models and even the current distribution. In the RCP8.5 scenario, the area of suitable habitats declines. The decline in the CCSM4-8.5 model is less than the other models. Coverage of the protected areas in different scenarios showed that in the RCP2.6 scenario, a larger range of suitable habitats is covered by the protected areas. These areas, under the RCP8.5 scenario, will keep a downtrend. The highest coverage of the protected areas was achieved by the CCSM4-2.6 model and the lowest by MRI-CGCM3-8.5.

### 3.3. Electric circuit theory

Fig 5 shows the outputs from implementing the circuit theory among the protected areas. The value of each pixel in this figure indicates the flow of the current passing through that pixel. In other words, it indicates the probability of the species passing from one protected area to another.

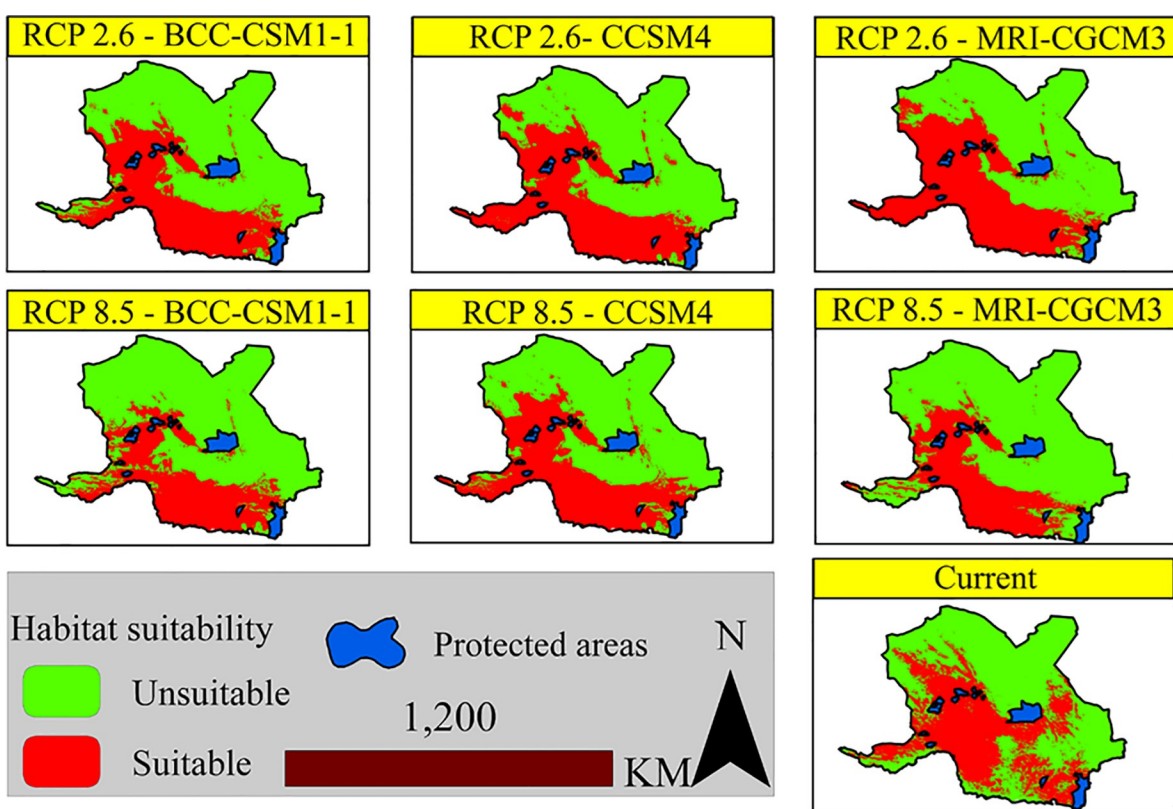

**Fig 4. Suitable/unsuitable habitat suitability maps of Asian black bears under the different climate change scenarios.**

**Table 5. Average height and area of the suitable habitats under the climate scenarios.**

| Scenarios | Area(sq Km) | % of Total Border | Inside PAs(sq Km) | Outside PAs(sq Km) |
|---|---|---|---|---|
| BCC-CSM1-2.6 | 161742.87 | 37.87 | 7332.03 | 154410.83 |
| CCSM4- 2.6 | 179007.53 | 41.91 | 8889.72 | 170117.80 |
| MRI-CGCM3-2.6 | 189731.22 | 44.42 | 7493.69 | 182237.52 |
| BCC-CSM1-8.5 | 135186.99 | 31.65 | 7923.76 | 127263.22 |
| CCSM4-8.5 | 170400.11 | 39.20 | 6893.97 | 163506.13 |
| MRI-CGCM3-8.5 | 126455.49 | 29.61 | 5577.43 | 120878.05 |
| Current | 174001.90 | 40.74 | 10337.50 | 163664.38 |

Blue and pink colors show less flow (less probability of connectivity) and gray shows higher flow (higher probability of connectivity). In all scenarios, the pathways are located at the heights of the three provinces. In the RCP2.6 scenario, the highest current flow is 11.88 amps obtained in the CCSM4 model. Higher values of amps indicate a higher current intensity and therefore a greater likelihood of connectivity between protected areas.

In the BCC-CSM1 and MRI-CGCM3 models, no current flow exists between the protected areas of Gando and Puzak.

The current connection is also established in the CCSM4 model, which due to its low width along the path, is considered as a pinch point. Under the RCP 8.5 scenario, the maximum current is 11.39 amps belonging to the MRI-CGCM3 model.

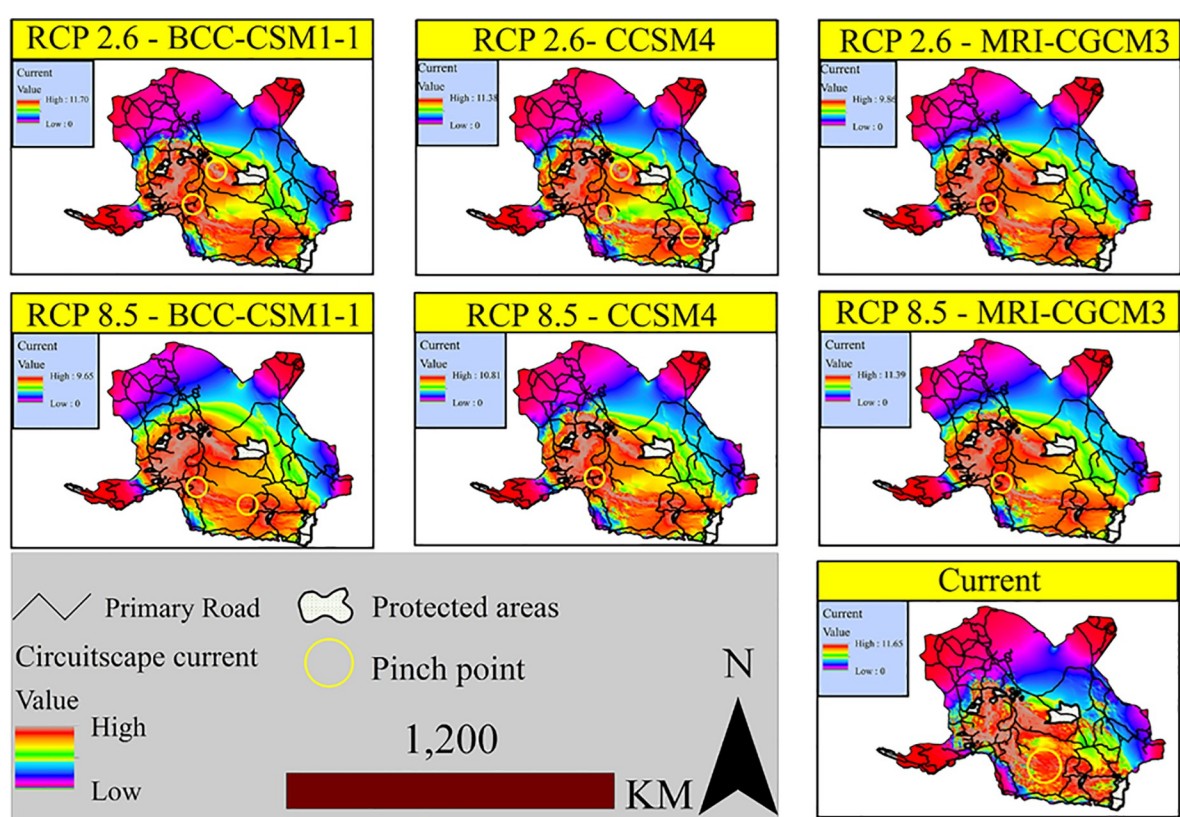

**Fig 5. Current flow between the present protected areas under the climate change scenarios.**

Compared to the models of RCP2.6 scenario, in the RCP8.5 scenario, the current flow is narrowed along the connection pathway of the protected areas in the Sistan-Baluchestan province to Hormozgan; however, the intensity of the flow is increased in the southern parts of Kerman Province and northern Hormozgan. The movement pinch points are currently located in the western part of Kahnooj in Kerman Province, and Chah Hashem Plain in Delgan City, Sistan-Baluchestan Province. The height of the Chah Hashem Plain is between 300 and 400 meters above sea level. The southern part has mountains with an altitude range of 1000 to 1500 meters, through which non-continuous current passes and in the northwest, it is connected to Kahnooj Heights and in the northern part to Kuh-e Jebal Barez. The movement pinch points under the different climate change scenarios and compared to the present state, have almost a definite location mainly found in the western highlands of Kahnooj towards the Fareghan and Khvoshkuh Mountains. This distance is surrounded by a low plain with various land uses included. Passing through these pinch points, the heights of Kuh-e Genu, Kuh-e Boz, and Kuh-e Ja'in will appear, all with high intensity of movement.

### 3.4. Influential variables

Table 6 shows the outputs from the random forest model. In this table, the importance of habitat variables is shown separately for the presence and pseudo-absence data. It also provides the overall importance of the mentioned variables based on the mean decrease in the accuracy and Gini coefficient. According to the results, the bio19, bio17, and bio7 variables had a greater effect on the habitat suitability, while the effect of habitat fragmentation index was recognized to be the least.

## 4. Discussion

Conservation measures for different wildlife species should be based on basic knowledge and awareness of species' biological needs and conservation options [86]. Habitat suitability models have helped to implement these measures by recognizing the species' needs and distribution. This study was conducted to investigate the connectivity of the habitats of Asian black bears (*Ursus thibetanus gedrosianus*) in southern and southeastern Iran.

**Table 6. Random forest model and the estimates on the effects of influential variables.**

| Variable | Absent | Present | Deceasing accuracy | Decreasing Gini |
|---|---|---|---|---|
| Altitude | 10.38 | 4.05 | 10.38 | 3.17 |
| Distance from garden | 5.35 | 10.23 | 10.84 | 3.02 |
| Distance from main road | 0.25 | 9.52 | 7.85 | 1.83 |
| Distance from streams | 1.81 | 0.97 | 0.14 | 0.61 |
| Distance from Human Statement | 3.93 | 9.38 | 10.12 | 2.24 |
| Landscape fragmentation | 0.89 | 0.81 | 1.14 | 0.70 |
| Cluster Hill shade of Elevation | 10.64 | 4.49 | 10.87 | 3.24 |
| NDVI | 7.84 | 12.49 | 14.00 | 7.63 |
| Bio3 | 13.57 | 14.59 | 18.47 | 5.11 |
| Bio7 | 13.25 | 19.36 | 19.68 | 8.50 |
| Bio11 | 10.30 | 7.29 | 12.51 | 2.82 |
| Bio15 | 17.70 | 14.04 | 19.51 | 6.67 |
| Bio17 | 14.88 | 25.09 | 24.83 | 10.10 |
| Bio18 | 13.13 | 13.49 | 16.77 | 6.67 |
| Bio19 | 11.51 | 22.41 | 23.21 | 11.72 |

In addition to the climatic variables, this study also used topographic, vegetation, and elevation variables. Modeling was performed using four models MaxEnt, GLM, BP-ANN, and SVM, which were finally integrated.

The validity of the model was confirmed based on AUC and TSS metrics (Table 3). According to the stacked binary prediction, the Ensemble modeling approach achieved a sensitivity of 98%, which means identifying 98% of the black bear's presence points after applying the threshold. The specificity value, in this model, was 0.93, which shows that the threshold could be able to separate pseudo-absence points with an accuracy of 93%. This is a considerable accuracy for the classification. The outputs, in terms of coverage, include most of the species habitats in the provinces of Sistan-Baluchestan ([49, 51], Hormozgan [57], and Kerman [46]. The random forest method showed that climatic variables (BIO19), precipitation of coldest quarter, precipitation of driest quarter (BIO17), and temperature annual range (BIO7) have the greatest impact on the suitability of the habitats. Unlike the high values of bio19 and bio17, which are desirable for the species, the black bear avoids high levels of Bio7. The bears avoid temperatures above 40°C and take refuge in caves at highlands [46]. In a study by Almasieh et al. (2016) on the habitat status of this species using the maximum entropy method, it was found that elevation and Bio12 were the most important habitat variables. However, elevation in this study did not have a significant effect on the model outputs. In the study conducted by Farashi and Erfani (2018), annual precipitation had the greatest effect on the modeling process. Accordingly, it seems that in a broad-scale (bioclimatic) approach, habitat suitability has a positive relationship with precipitation and a negative with temperature. Although this study did not recognize altitude as an important variable, this factor, by influencing temperature and precipitation gradients, can affect the suitability of habitats. As the altitude increases from 300 m to 2,500 m, the habitat suitability for the species increases. This is in line with the findings by Farashi and Erfani (2018).

In a study by Bista and Aryal (2013) on Asian black bears (*Ursus thibetanus*) in central Nepal, the optimal height for the species was reported to be between 1600 m and 3200 m. In another study, Ali et al. (2017) reported the elevation of between 2,500 and 3,000 meters for *Ursus thibetanus* in the eastern Himalayas in Pakistan as the height of presence, which is lower than that found for the subspecies of Asian black bears in Iran.

After climatic variables, NDVI would be of utmost importance. High values of NDVI are desirable for the species. The presence of this species in Jebal Barez and Delfard forests in the south of Kerman also confirms the importance of NDVI for the species. Montane woodlands in southern Kerman are considered to be among rich and unique habitats for this species as it provides food for the bears in all seasons of the year. Asian black bears in Manaslu Conservation in Nepal also tend to the oak forest [56]. Areas with less vegetation density mainly include the habitats in Sistan-Baluchestan (Nikshahr area) as well as the east and north of Hormozgan Province [51]. This is in agreement with the findings reported by Bista et al. (2018). They introduced the conifer forest as the most important habitat type for black bears [54]. Escobar et al. (2015) found the same results and reported a decline of 10% in the suitable areas for Asian black bears due to habitat loss [123].

Distance from gardens as one of the most important variables in this study was found to be more important than "height" and "distance from residential areas" (Table 6). The tendency to these areas is due to the role that orchards play in feeding the black bears. The results of a study conducted by Fahimi et al. (2011) on this species in Dalfard-Dehbakri, Kerman, also showed that black bears are more interested in gardens. Food supply is a key factor in the distribution and home range of bears [124]. A study conducted by Garshelis and Steinmetz (2008) on Asian black bears showed that food availability could be effective in determining the home range of the species. Although the species"high willingness to the presence in farmlands"

can be a source of conflict, with regards to the home range [48] and the potential navigation ability [125] of black bears, the situation would be exacerbated when much of the potential habitats for the species is outside the boundaries of protected areas (Table 5). Most species-related conflicts occur during sunset, which may be due to human presence and insecurity during the daytime [123]. In areas where human presence, along with access to food, is a limiting factor, the behavior of the species may be changed to nucturnality [126]. There are reports of attacks on humans in Dalfard (Jiroft, Kerman Province) and Gol Gol Heights (Roudan County, Hormozgan Province). Numerous reports of such conflicts are available in Pakistan [53] and Bhutan [127].

The results of different climate change scenarios showed that the area of suitable habitats in the RCP2.6 scenario was higher than that of the RCP8.5 scenario, revealing a negative impact of climate change on the species distribution. Under the severe scenarios, the area of suitable habitats is declined in the eastern and western regions (Table 5). The negative effect of climate change on the species has been emphasized in other studies [2]. According to the results of circuit theory, the Chah Hashem Plain and the southwestern plain of Kahnooj are the current and future movement pinch points of the species, where are characterized by low altitude compared to the adjacent areas. The situation of these movement bottlenecks is different from each other; which somehow reflects the effect of climate change on the species movement. Since wildlife species will seek areas with similar climatic conditions of the original habitats [128], the protected areas should be selected along the identified corridors in a way to include the caves of the highlands, this could be an appropriate strategy. The quality of the movements between the protected areas may be affected by the passage through the main roads (Fig 5). As such, mining operations in the Henza Mountains in the Bahr_Aseman area have led to insecurity and the exodus of bears from the area and road accidents. The black bears can be considered a kind of umbrella species; by protecting it, many plants and animals specific to the habitats of southern and southeastern Iran will be preserved, as well. Seed germination in scats is recorded within the distribution range of black bears for date palm (*Phoenix dactylifera*) [51, 52]. In the habitats of this species in Shushin area in Balochistan, the presence of the species such as wild goat (*Capra aegagrus*), wild sheep (*Ovis vignei*), Persian leopard (*Panthera pardus*), Indian crested porcupine (*Hystrix indica*) and wild boar (*Sus scrofa*) has been recorded [129]. These species are not the only recorded ones. In the protected area of Bazman, the presence of chinkara (*Gazella bennettii*), bustard (*Chlamydotis macqueenii*), and sand cat (*Felis margarita*) has also been confirmed.

## 5. Conclusions

To provide gene flow and population connectivity, this species needs connections in the whole landscape; that way, it can have access to seasonal food and the new population can move towards suitable habitat patches. To study the habitat connections and to identify migration corridors for the Asian black bears in the areas under study, circuit theory was used.

Bahr Aseman Protected Area, Zaryab Wildlife Refuge, Pozak Protected Area, Geno Biosphere Reserve, and Kouhbaz Protected Area are the most important habitats of Asian black bears. The species has occasionally been spotted in Bazman Area and rarely observed in Gando Protected Area. Ruchoon Wildlife Refuge and Khabr National Park used to have bears in the past, but there is no bear in the park currently. Kouh Shir Protected Area lacks bear species while, in terms of habitat conditions, it is similar to the Sang-e Mes Protected Area, where is considered as the main habitat of the present species. These areas could be targeted for restoration to bring the species back to them. As the outputs show, the highest flow was found in Bahr Aseman Protected Area, Zaryab Wildlife Refuge, and Sang-e Mes Protected Area. Thus,

Kerman Province compared to Sistan-Baluchestan, and Hormozgan Provinces, had more desirable conditions for the black bears.

## Supporting information

**S1 File.**
(ZIP)

## Author Contributions

**Investigation:** Maryam Morovati.

**Validation:** Maryam Morovati.

**Writing – original draft:** Maryam Morovati.

**Writing – review & editing:** Peyman Karami, Fatemeh Bahadori Amjas.

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
