## [Decision Letter · Decision Letter 0]

8 May 2020

PONE-D-20-07352

Identifying connections between habitat patches and modeling habitat suitability under the influence of climate change (a case study of Asiatic black bear in the Iranian landscape)

PLOS ONE

Dear Dr Morovati,

Thank you for submitting your manuscript to PLOS ONE. After careful consideration, we feel that it has merit but does not fully meet PLOS ONE’s publication criteria as it currently stands. Therefore, we invite you to submit a revised version of the manuscript that addresses the points raised during the review process.

We would appreciate receiving your revised manuscript by Jun 21 2020 11:59PM. To enhance the reproducibility of your results, we recommend that if applicable you deposit your laboratory protocols in protocols.io, where a protocol can be assigned its own identifier (DOI) such that it can be cited independently in the future. For instructions see: http://journals.plos.org/plosone/s/submission-guidelines#loc-laboratory-protocols

We look forward to receiving your revised manuscript.

Kind regards,

Lyi Mingyang, Ph.D.

Academic Editor

PLOS ONE

Additional Editor Comments:

Dear authors,

thank you for the submission of your interesting manuscript to PLOSONE.

The two reviewers find the content rather compelling while indicating critical points to address. The reviewers provided very useful suggestions to improve the overall clarity of your study as well as the quality of your analysis. The suggestions of the reviewers look feasible to me and I believe you will be able to address them. Thus, please take care to do a full revision of your manuscript according to all reviewers’ comments. Improvements based on reviewers’ comments will be crucial for acceptance.

Best regards,

LM

Journal Requirements:

2. We note that Figures 1, 4 and 5 in your submission contain map images which may be copyrighted. All PLOS content is published under the Creative Commons Attribution License (CC BY 4.0), which means that the manuscript, images, and Supporting Information files will be freely available online, and any third party is permitted to access, download, copy, distribute, and use these materials in any way, even commercially, with proper attribution. For these reasons, we cannot publish previously copyrighted maps or satellite images created using proprietary data, such as Google software (Google Maps, Street View, and Earth). For more information, see our copyright guidelines: http://journals.plos.org/plosone/s/licenses-and-copyright.

2.1.You may seek permission from the original copyright holder of Figures 1, 4 and 5 to publish the content specifically under the CC BY 4.0 license.

2.2. If you are unable to obtain permission from the original copyright holder to publish these figures under the CC BY 4.0 license or if the copyright holder’s requirements are incompatible with the CC BY 4.0 license, please either i) remove the figure or ii) supply a replacement figure that complies with the CC BY 4.0 license. Please check copyright information on all replacement figures and update the figure caption with source information. If applicable, please specify in the figure caption text when a figure is similar but not identical to the original image and is therefore for illustrative purposes only.

3. Thank you for stating the following financial disclosure: "No"

5. Please upload a new copy of Figure 2 as the detail is not clear. Please follow the link for more information: https://blogs.plos.org/plos/2019/06/looking-good-tips-for-creating-your-plos-figures-graphics/

Reviewers' comments:

Reviewer's Responses to Questions

**Comments to the Author**

1. Is the manuscript technically sound, and do the data support the conclusions?

Reviewer #1: Yes

Reviewer #2: No

2. Has the statistical analysis been performed appropriately and rigorously? 

Reviewer #1: Yes

Reviewer #2: No

3. Have the authors made all data underlying the findings in their manuscript fully available?

Reviewer #1: Yes

Reviewer #2: Yes

4. Is the manuscript presented in an intelligible fashion and written in standard English?

Reviewer #1: No

Reviewer #2: No

5. Review Comments to the Author

Reviewer #1: General comments:

The manuscript “Identifying connections between habitat patches and modeling habitat suitability under the influence of climate change (a case study of Asiatic black bear in the Iranian landscape)”, submitted to PLOSONE by Morovati et al., is a well done and very interesting study. This ms provides interesting information but it needs to be improved. I think that some references should be added in some specific point of the text. Moreover, this manuscript should be absolutely revised by a English Native Speaker. Please, see below my specific comments.

Specific comments:

Line 2: Please, change the word “Asiatic black bear” with the scientific name “Ursus thibetanus gedrosianus” . Check it in all the part of manuscript

Lines 56 – 58: Please, rewrite this sentence. It is unintelligible

Line 85: Please, arrange the keywords in alphabetic order

Line 59: Add the scientific after “Asiatic black bear”

Lines 71 – 73: Is it normal all these values = zero?

Lines 96 – 98: I would like to suggest to add in the introduction also some short sentences on this other recent study carried out on the effect of climate change on jerboas species in Iran

Mohammadi, S., et al. (2019). Modelling current and future potential distributions of two desert jerboas under climate change in Iran. Ecological Informatics, 52, 7-13.

Lines 117 – 119: I think that you should add some references as examples to support this your sentence “According to them, human interventions, such as farming, urban development, and infrastructure construction for transportation purposes are among are main reasons for fragmentation of natural landscape.” I would like to suggest:

Smeraldo, S., et al., (2020). Modelling risks posed by wind turbines and power lines to soaring birds: the black stork (Ciconia nigra) in Italy as a case study. Biodiversity and Conservation, 29, 1959-1976.

Jensen, A. M., et al., (2019). Landscape effects on the contemporary genetic structure of ruffed grouse (Bonasa umbellus) populations. Ecology and Evolution, 9(10), 5572-5592.

Lines 141 – 143: Please, rewrite this sentence in this way and add the following references to support it: “Such a modeling also facilitates achieving some management goals, including conservation of endangered plants and animal populations, prediction of the prevalence of infectious diseases, the early detection of invasive species and protection of biodiversity (Bertolino et al., 2020; Bosch et al., 2019; Thompson and Brooks-Pollock, 2019).

Bertolino S., et al., (2020). Spatially-explicit models as tools for implementing effective management strategies for invasive alien mammals. Mammal Review, 50, 187-199.

Thompson, R. N., & Brooks-Pollock, E. (2019). Detection, forecasting and control of infectious disease epidemics: modelling outbreaks in humans, animals and plants. Phil. Trans. R. Soc. B37420190038, http://doi.org/10.1098/rstb.2019.0038

Bosch, M., et al., (2019). New conservation viewpoints when plants are viewed at one level higher. Integration of phylogeographic structure, niche modeling and genetic diversity in conservation planning of W Mediterranean larkspurs. Global ecology and conservation, 18, e00580.

Line 161: Please, start a new line

Line 251: Please, delete the underline

Lines 262 – 263: I think that you should add some references as examples to support this your sentence “The Maxent method offers reliable answers, compared to other habitat utility models, with fewer presence points.” I would like to suggest:

Ancillotto, L., et al. (2019). The Balkan long-eared bat (Plecotus kolombatovici) occurs in Italy – first confirmed record and potential distribution. Mammalian Biology, 96: 61-67.

Acharya, B. K., et al., (2019). Mapping Environmental Suitability of Scrub Typhus in Nepal Using MaxEnt and Random Forest Models. International Journal of Environmental Research and Public Health, 16(23), 4845.

Lines 263 – 265: Please, add more information regarding the Maxent settings used in this study (e.g., number of iterations, run type, etc…)

Line 269: “Jackknife analysis” instead of “Jack Nayef analysis”. Check it in all the manuscript

Lines 266 – 269: Please, note that only AUC is not sufficient to validate your models. You should use also the True Skill Statistics (TSS) method

Allouche, O., Tsoar, A., & Kadmon, R. (2006). Assessing the accuracy of species distribution models: prevalence, kappa and the true skill statistic (TSS). Journal of applied ecology, 43(6), 1223-1232.

Lines 282 – 295: I think that you should add some references to support of this methods. I would like to suggest:

Finch, D., et al., (2020). Modelling the functional connectivity of landscapes for greater horseshoe bats Rhinolophus ferrumequinum at a local scale. Landscape Ecology, 35(3), 577-589.

Freeman, B., et al., (2019). Modeling endangered mammal species distributions and forest connectivity across the humid Upper Guinea lowland rainforest of West Africa. Biodiversity and conservation, 28(3), 671-685.

Parrott, L., et al., (2019). Planning for ecological connectivity across scales of governance in a multifunctional regional landscape. Ecosystems and People, 15(1), 204-213.

Lines 326 – 332: Please, add the list of these variable in a Table

Line 348: Please, split this paragraph in two (Results) and (Discussion)

Figure 3: Please, slit it in two figure (a-f) and (g-l)

Line 510: Please, expand this section of the manuscript and discuss your results with other studies

Reviewer #2: The study models current and future niche of Asiatic black bear in the Iranian landscape. Perks of this study include modeling not only current but also future distribution, developing maps under several climate scenarios, and identifying potential suitable corridors. While I certainly feel as though this could be an interesting analysis, as the manuscript stands there are enough methodological and modeling concerns that would require a complete reconstruction of the analysis and manuscript. Indeed, the introduction is a bit convoluted and difficult to follow, and the entire manuscript is in need of thorough review for grammar (e.g., tense agreements), punctuation, and general continuity and flow. Of greater concern, however, are the outdated methods implemented. While I realize that this study is a primarily a model-based assessment, and as such, there is a higher bar when it comes to the appropriate methodological approaches. As they are communicated presently, I have little confidence that the best models possible were generated and doubt that they provide any additional or useful insight about the study species. Because of these methodological issues, and subsequent lack of confidence in the models produced, I have provided minimal commentary on the results and discussion sections and focus almost exclusively on methods.

6. PLOS authors have the option to publish the peer review history of their article (what does this mean?). If published, this will include your full peer review and any attached files.

Reviewer #1: No

Reviewer #2: Yes: Nader Habibzadeh

---

## [Author Response · Author response to Decision Letter 0]

20 Jul 2020

Answer to :Reviewer #1:

Thank you very much for your suggestions and criticisms

The modeling and body of this article have been rewritten.

Model input variables, models used, climate change scenarios, and modeling of habitat connections have completely changed.

Reviewer #1: General comments:

The manuscript “Identifying connections between habitat patches and modeling habitat suitability under the influence of climate change (a case study of Asiatic black bear in the Iranian landscape)”, submitted to PLOSONE by Morovati et al., is a well done and very interesting study. This ms provides interesting information but it needs to be improved. I think that some references should be added in some specific point of the text. Moreover, this manuscript should be absolutely revised by a English Native Speaker. Please, see below my specific comments.

Specific comments:

Line 2: Please, change the word “Asiatic black bear” with the scientific name “Ursus thibetanus gedrosianus” . Check it in all the part of manuscript

Answer to the Reviewer:

It was done

Lines 56 – 58: Please, rewrite this sentence. It is unintelligible

Answer to the Reviewer:

Dear Referee, The introduction, materials, methods and results of this study have been completely revised.

Line 85: Please, arrange the keywords in alphabetic order

Answer to the Reviewer:

It was done

Line 59: Add the scientific after “Asiatic black bear”

Answer to the Reviewer:

Introduction has changed, however, was observed in all text introduction

Lines 71 – 73: Is it normal all these values = zero?

Answer to the Reviewer:

This part has also been changed and using the random forest method, the importance of the influential variables in the study has been determined.

Lines 96 – 98: I would like to suggest to add in the introduction also some short sentences on this other recent study carried out on the effect of climate change on jerboas species in Iran

Mohammadi, S., et al. (2019). Modelling current and future potential distributions of two desert jerboas under climate change in Iran. Ecological Informatics, 52, 7-13.

Answer to the Reviewer:

The introduction has been changed and a reference has been made to this study.

Lines 117 – 119: I think that you should add some references as examples to support this your sentence “According to them, human interventions, such as farming, urban development, and infrastructure construction for transportation purposes are among are main reasons for fragmentation of natural landscape.” I would like to suggest:

Smeraldo, S., et al., (2020). Modelling risks posed by wind turbines and power lines to soaring birds: the black stork (Ciconia nigra) in Italy as a case study. Biodiversity and Conservation, 29, 1959-1976.

Jensen, A. M., et al., (2019). Landscape effects on the contemporary genetic structure of ruffed grouse (Bonasa umbellus) populations. Ecology and Evolution, 9(10), 5572-5592.

Answer to the Reviewer:

The introduction has changed and related sentences have been used in the overall review.

Lines 141 – 143: Please, rewrite this sentence in this way and add the following references to support it: “Such a modeling also facilitates achieving some management goals, including conservation of endangered plants and animal populations, prediction of the prevalence of infectious diseases, the early detection of invasive species and protection of biodiversity (Bertolino et al., 2020; Bosch et al., 2019; Thompson and Brooks-Pollock, 2019).

Bertolino S., et al., (2020). Spatially-explicit models as tools for implementing effective management strategies for invasive alien mammals. Mammal Review, 50, 187-199.

Thompson, R. N., & Brooks-Pollock, E. (2019). Detection, forecasting and control of infectious disease epidemics: modelling outbreaks in humans, animals and plants. Phil. Trans. R. Soc. B37420190038, http://doi.org/10.1098/rstb.2019.0038

Bosch, M., et al., (2019). New conservation viewpoints when plants are viewed at one level higher. Integration of phylogeographic structure, niche modeling and genetic diversity in conservation planning of W Mediterranean larkspurs. Global ecology and conservation, 18, e00580.

Answer to the Reviewer:

The introduction has changed and related sentences have been used in the overall review.

Line 161: Please, start a new line

Answer to the Reviewer:

The introduction has changed and related sentences have been used in the overall review.

Line 251: Please, delete the underline

Answer to the Reviewer:

In addition to modeling using the maximum entropy model (MaxEnt), other methods of modeling such as Generalized Linear Models (GLM), Support vector machines (SVM), artificial neural network (BP-ANN) and consensus model (Ensemble) It was also used in the modeling process

Lines 262 – 263: I think that you should add some references as examples to support this your sentence “The Maxent method offers reliable answers, compared to other habitat utility models, with fewer presence points.” I would like to suggest:

Ancillotto, L., et al. (2019). The Balkan long-eared bat (Plecotus kolombatovici) occurs in Italy – first confirmed record and potential distribution. Mammalian Biology, 96: 61-67.

Acharya, B. K., et al., (2019). Mapping Environmental Suitability of Scrub Typhus in Nepal Using MaxEnt and Random Forest Models. International Journal of Environmental Research and Public Health, 16(23), 4845.

Answer to the Reviewer:

The models used in this study have changed and the presence / absence models have entered modeling.

Lines 263 – 265: Please, add more information regarding the Maxent settings used in this study (e.g., number of iterations, run type, etc…)

Answer to the Reviewer:

In the revised version, the maximum entropy model was used along with other desirability models, and the number of pseudo-absence points was considered to be twice the total number of presence points, and modeling was performed.

Line 269: “Jackknife analysis” instead of “Jack Nayef analysis”. Check it in all the manuscript

Answer to the Reviewer:

Random forest method was used to investigate the importance of variables.

Lines 266 – 269: Please, note that only AUC is not sufficient to validate your models. You should use also the True Skill Statistics (TSS) method

Allouche, O., Tsoar, A., & Kadmon, R. (2006). Assessing the accuracy of species distribution models: prevalence, kappa and the true skill statistic (TSS). Journal of applied ecology, 43(6), 1223-1232.

Answer to the Reviewer:

In the new version, AUC and TSS indicators were used as threshold non-dependent indicators for modeling evaluation.

Lines 282 – 295: I think that you should add some references to support of this methods. I would like to suggest:

Finch, D., et al., (2020). Modelling the functional connectivity of landscapes for greater horseshoe bats Rhinolophus ferrumequinum at a local scale. Landscape Ecology, 35(3), 577-589.

Freeman, B., et al., (2019). Modeling endangered mammal species distributions and forest connectivity across the humid Upper Guinea lowland rainforest of West Africa. Biodiversity and conservation, 28(3), 671-685.

.

Parrott, L., et al., (2019). Planning for ecological connectivity across scales of governance in a multifunctional regional landscape. Ecosystems and People, 15(1), 204-213.

Answer to the Reviewer:

In the new version, this part has been changed. And references have been added to this section

Lines 326 – 332: Please, add the list of these variable in a Table

Answer to the Reviewer:

In the new version of the article, the variables used in the materials section and study methods are mentioned

Line 348: Please, split this paragraph in two (Results) and (Discussion)

Answer to the Reviewer:

In the new version, all sections of the article are separated

Figure 3: Please, slit it in two figure (a-f) and (g-l)

Answer to the Reviewer:

In the revised version, all images are listed separately by execution scenarios

Line 510: Please, expand this section of the manuscript and discuss your results with other studies

Answer to the Reviewer:

In discussing the revised version and compare it with other findings was performed.

Answer to :Reviewer #2:

Many thanks to the esteemed judge, it is noted that all sections of the article were revised based on the opinions of the esteemed judges.

Reviewer #2

Reviewer #2: The study models current and future niche of Asiatic black bear in the Iranian landscape. Perks of this study include modeling not only current but also future distribution, developing maps under several climate scenarios, and identifying potential suitable corridors. While I certainly feel as though this could be an interesting analysis, as the manuscript stands there are enough methodological and modeling concerns that would require a complete reconstruction of the analysis and manuscript. Indeed, the introduction is a bit convoluted and difficult to follow, and the entire manuscript is in need of thorough review for grammar (e.g., tense agreements), punctuation, and general continuity and flow. Of greater concern, however, are the outdated methods implemented. While I realize that this study is a primarily a model-based assessment, and as such, there is a higher bar when it comes to the appropriate methodological approaches. As they are communicated presently, I have little confidence that the best models possible were generated and doubt that they provide any additional or useful insight about the study species. Because of these methodological issues, and subsequent lack of confidence in the models produced, I have provided minimal commentary on the results and discussion sections and focus almost exclusively on methods.

Answer to Reviewer

Dear judge, in accordance with your comments

A complete review of the study method was performed. This review includes a review of the introduction, materials and methods, results and findings of the study.

The most important and prominent changes that have been made in order to address the concerns of the esteemed arbitrator in the section of materials and working methods are as follows:

1-Using appropriate non-attendance methods and preparing pseudo-absence points

2-Using the methods of presence / absence and Hamadi model

3-Calculation of dependent indices (Kappa, sensitivity and specificity ..) and non-dependent thresholds (TSS, AUC)

4-Using different climate change scenarios in the modeling process

5-Using random forest method to evaluate the importance of variables

In line with these changes, the results and discussion of the article have also changed, which we hope will allay your concerns about the proper implementation of the approach.

---

## [Decision Letter · Decision Letter 1]

18 Aug 2020

PONE-D-20-07352R1

Accessing Habitat Suitability and Connectivity for the Easternmost Population of Asian Black Bear (Ursus thibetanus gedrosianus, Blanford, 1877) based on Climate Changes Scenarios in Iran

PLOS ONE

Dear Dr. Morovati,

Thank you for submitting your manuscript to PLOS ONE. After careful consideration, we feel that it has merit but does not fully meet PLOS ONE’s publication criteria as it currently stands. Therefore, we invite you to submit a revised version of the manuscript that addresses the points raised during the review process.

We look forward to receiving your revised manuscript.

Kind regards,

Lyi Mingyang, Ph.D.

Academic Editor

PLOS ONE

Additional Editor Comments (if provided):

Dear Authors,

the two reviewers asked to you other important revisions in order to improve your manuscript.

I have read your paper and I agree with them that currently your manuscript is not stil suitable for the publication in PLOSONE.

I suggest to you to pay more attention to all the requests.

Best regards,

LM

Reviewers' comments:

Reviewer's Responses to Questions

**Comments to the Author**

1. If the authors have adequately addressed your comments raised in a previous round of review and you feel that this manuscript is now acceptable for publication, you may indicate that here to bypass the “Comments to the Author” section, enter your conflict of interest statement in the “Confidential to Editor” section, and submit your "Accept" recommendation.

Reviewer #1: (No Response)

Reviewer #2: (No Response)

2. Is the manuscript technically sound, and do the data support the conclusions?

Reviewer #1: Yes

Reviewer #2: (No Response)

3. Has the statistical analysis been performed appropriately and rigorously? 

Reviewer #1: Yes

Reviewer #2: (No Response)

4. Have the authors made all data underlying the findings in their manuscript fully available?

Reviewer #1: Yes

Reviewer #2: (No Response)

5. Is the manuscript presented in an intelligible fashion and written in standard English?

Reviewer #1: No

Reviewer #2: (No Response)

6. Review Comments to the Author

Reviewer #1: General comments:

The manuscript “Accessing Habitat Suitability and Connectivity for the Easternmost Population of Asian Black Bear (Ursus thibetanus gedrosianus, Blanford, 1877) based on Climate Changes Scenarios in Iran”, submitted to PLOSONE by Morovati et al., has definitely improved after the first revision, however, the authors have yet to work long before that this paper can be definitively accepted. This ms provides interesting information but it needs to be improved. I think that some references should be added in some specific point of the text. Moreover, this manuscript should be absolutely revise by a English Native Speaker. Please, see below my specific comments.

Specific comments:

Line 30: To use the acronyms (e.g., GLM) only if you will use them another time in the abstract. Please, check the acronyms in all the abstract.

Line 64: [4] not superscript.

Lines 81 – 82: I think that you should add some references as examples to support this your sentence “Models with the ability to predict the suitability of wildlife habitats on a large scale can be useful for wildlife managers.” I would like to suggest:

Bertolino S., et al., (2020). Spatially-explicit models as tools for implementing effective management strategies for invasive alien mammals. Mammal Review, 50, 187-199.

Pauli, B. P., et al. (2019). Human habitat selection: using tools from wildlife ecology to predict recreation in natural landscapes. Natural Areas Journal, 39(2), 142-149.

Lines 83 – 84: I think that you should add some references as examples to support this your sentence “For the protection of an important species, it is of critical importance to identify its needs as well as habitat constraints, and degradation factors.” I would like to suggest:

Smeraldo, S., et al., (2020). Modelling risks posed by wind turbines and power lines to soaring birds: the black stork (Ciconia nigra) in Italy as a case study. Biodiversity and Conservation, 29, 1959-1976.

Andrade-Díaz, M. S., et al., (2019). Expansion of the agricultural frontier in the largest South American Dry Forest: Identifying priority conservation areas for snakes before everything is lost. PloS one, 14(9), e0221901.

Lines 182 – 183: I think that you should add some references as examples to support this your sentence “After identifying the threshold, the metrics of sensitivity, specificity, correct classification, and miss classification were used to assess the power of the threshold [3].” I would like to suggest:

Ancillotto, L., et al. (2020). An African bat in Europe, Plecotus gaisleri: Biogeographic and ecological insights from molecular taxonomy and Species Distribution Models. Ecology and Evolution, 10, 5785-5800.

Lines 209 – 210: This figure should be moved in the Results. To delete 0’0” from the figure.

Line 327: Why did you colour some number in grey?

Reviewer #2: (No Response)

7. PLOS authors have the option to publish the peer review history of their article (what does this mean?). If published, this will include your full peer review and any attached files.

Reviewer #1: No

Reviewer #2: No

---

## [Author Response · Author response to Decision Letter 1]

24 Oct 2020

Ms.No.PONE-D-20-07352

There were several parts of the manuscript (MS) that were confusing and unfocused despite reviewer#2 highlighted several major issues, especially those pertaining to methodological and modeling concerns including the delimitation of ‘model calibration region’ and ‘Model uncertainty’. As the MS stands, much of the confusion seems to come from methods and results sections yet because the whole paper did not comprehensively follow reviewer’s comments. Some of my specific comments below may help, but in general, the authors should try to improve readability and clarity by thinking about the organization of the paper and the logical transitional flow between ideas.

Dear Editor,

We hereby appreciate valuables comments. The points were applied in the text-body and highlighted in green. Before answering getting started, let me provide you with some explanations about the first-round revisions. In the first round of revising the paper, objections and suggestions were made by the first and second Referees, the response to which required a major and complete revision of the article. According to the first-round revisions, the concerns of both Referees have been answered and applied in the paper, which generally included the following items:

1- All sections of the article such as abstract, introduction, materials and methods, results and discussion, and conclusion were revised completely.

2- Instead of using one model, several models have been applied.

3- Instead of using one climate change scenario, several different models under the RCP2.6 and RCP8.5 scenarios have been used.

4- The efficiency of the network of the protected areas in covering the suitable habitats has also been studied. 

The changes will be obvious if this version is compared to the previous one, as we invite to kindly do so.

Thank you again for the time spending to read the rebuttal letter.

Hope the revisions would be satisfactory enough to meet the satisfaction of honorable Editor and Referees.

Truly yours

The authors

SPECIFIC/MINOR ISSUES

Line 25: change ‘communications’ with ‘connectivity’

[Authors’ response]: the revision was applied.

Line 28: remove ‘For this purpose’ 

[Authors’ response]: the revision was applied.

Line 28: change ‘3 provinces’ to ‘provinces’

[Authors’ response]: the revision was applied.

Line 29: change ‘Sistan and Baluchestan’ to ‘Sistan-Baluchestan’. Follow this afterwards throughout the text.

[Authors’ response]: the revision was applied.

Line 29: remove ‘and then, corrected in terms of spatial autocorrelation’

[Authors’ response]: the revision was applied.

Lines 30-31: delete the abbreviations of modelling techniques

[Authors’ response]: the revision was applied.

Line 30: change ‘,3 machine learning models of’ to ‘ and 3 machine learning models including’

[Authors’ response]: the revision was applied.

Line 31: Ensemble model is not itself a unique modelling algorithm, it is just an approach to combine the results of formal habitat modelling algorithms. So, the authors should address how the ensemble model was undertaken to integrate the main habitat models as a separate sentence. 

[Authors’ response]: thank you very much for the comment. We used the term of “model” is referring for the following reasons:

Ensemble in ModEco is introduced as a model with a meta-algorithm:

“- Bootstrap aggregating (bagging) is a machine learning ensemble meta-algorithm to improve classification and regression models in terms of stability and classification accuracy. 

- AdaBoost, short for Adaptive Boosting, is a meta-algorithm, and can be used in conjunction with many other learning algorithms to improve their performance.” 

In general, depending on the type of model used, the ensemble can be a very simple algorithm or use mixed algorithms in its background for modeling. Ensemble is referred as “Model” in the software interface. Please take a look to the following print screen:

Figure 1: Ensemble model direction in ModEco software

There are many articles referring to Ensemble as a model:

Guo, Q. and Liu, Y., 2010. ModEco: an integrated software package for ecological niche modeling. Ecography, 33(4), pp.637-642.

- Kafaei, S., Akmali, V. and Sharifi, M., 2020. Using the Ensemble Modeling Approach to Predict the Potential Distribution of the Muscat Mouse-Tailed Bat, Rhinopoma muscatellum (Chiroptera: Rhinopomatidae), in Iran. Iranian Journal of Science and Technology, Transactions A: Science, pp.1-12.

- Lei, J., Chen, L. and Li, H., 2017. Using ensemble forecasting to examine how climate change promotes worldwide invasion of the golden apple snail (Pomacea canaliculata). Environmental Monitoring and Assessment, 189(8), p.404.

- Mugo, R. and Saitoh, S.I., 2020. Ensemble Modelling of Skipjack Tuna (Katsuwonus pelamis) Habitats in the Western North Pacific Using Satellite Remotely Sensed Data; a Comparative Analysis Using Machine-Learning Models. Remote Sensing, 12(16), p.2591.

Anyhow, according to the Referee’s comment, we modified the “Ensemble model” in the paper as Ensemble modeling approach.

Lines 32-34: I am not sure that I could understand well what the authors mean by how they generated the background points to undertake the habitat modelling. It would better not mention it here and go through to explain it in depth in the appropriate part of the MS. 

[Authors’ response]: In this study, pseudo-absence points were prepared based on the Ensemble approach and presence-only models, i.e. the presence-only models were considered, and then modeling was done using these models. Finally, the results of these models were integrated. The map obtained from the combination of these models was used to prepare pseudo-absence points. 

According to the referee’s comment, the sentence explaining this process was removed and replaced by a short sentence.

Line 36: remove ‘,which have the best response for Iran,’

[Authors’ response]: the sentence was deleted. 

Lines 39-40: rephrase as: ‘The true skill statistic was employed to convert the continuous suitability layers to binary suitable/unsuitable range maps to assess the effectiveness of the protected areas in the coverage of suitable habitats for the species.’ It should point out why the authors chose this evaluation metric to dichotomize the habitat suitability, not here but the next parts of the MS.

[Authors’ response]: the abstract was modified and the explanation was added to the MS where appropriate. 

Lines 40-41: better to rephrase as: ‘The random forest method was also used to predict the relationships between the presence probability of the species and the environmental predicators.’ But, it should be highlighted in the other sections of the MS not here the reasons behind selecting this model to evaluate the species response curves to the predictors. 

 [Authors’ response]: the abstract was modified as recommended and in the text-body, it was explained that this model was used due to its high efficiency in identifying the influential variables. 

Line 42: To be specific. which model(s)? Did all models perform well in discriminating suitable and unsuitable areas?

 [Authors’ response]: the mean was the presence/pseudo-absence models used in the modeling process as mentioned in the abstract. 

Line 42: what does mean by ‘successful in the implementation’?

 [Authors’ response]: the mean by "Successful in the implementation" was “the successful run of the used models as they showed acceptable values of sensitivity and specificity. A similar term is found in other examples such as the followings:

- Cao, Y., DeWalt, R.E., Robinson, J.L., Tweddale, T., Hinz, L. and Pessino, M., 2013. Using Maxent to model the historic distributions of stonefly species in Illinois streams: the effects of regularization and threshold selections. Ecological Modelling, 259, pp.30-39.

- Fukuda, S., De Baets, B., Mouton, A.M., Waegeman, W., Nakajima, J., Mukai, T., Hiramatsu, K. and Onikura, N., 2011. Effect of model formulation on the optimization of a genetic Takagi–Sugeno fuzzy system for fish habitat suitability evaluation. Ecological modelling, 222(8), pp.1401-1413.

Line 43: be specific in what bio17, … stands for? Then delete the words of bio17, … here.

 [Authors’ response]: the complete name of all variables was included in the abstract.

Lines 43-44: clarify what the authors mean by ‘high’ and ‘extreme’? 

 [Authors’ response]: This sentence is based on the results of modeling and distribution of suitable habitats of the species in the study area, which is also mentioned in the discussion and conclusion. However, this sentence was corrected in the abstract.

Line 45: To which habitat modelling techniques were the area of suitable habitats shown in parentheses associated?

 [Authors’ response]: The area calculation was done only based upon the Ensemble models because Ensemble was used to generalize (prediction) the modeling results.

Line 46: replace ‘higher’ with ‘larger’ and ‘the current’ with ‘the current distribution’.

 [Authors’ response]: The revisions were applied as recommended. 

Line 46: ‘all the models run under…’. I am completely confused about why the authors applied the ensemble model regardless they are interested in addressing the changes of the species habitat suitability through the application of every single model? by considering that the main idea behind using the ensemble modelling is to deal with the uncertainties inherited from each modelling algorithm. 

[Authors’ response]: In this sentence, the authors meant that in each model a decrease was observed in the full coverage of suitable habitats of the species by the protected areas. This sentence was presented here to say that in all of the models, the network of the protected areas failed to fully cover the habitats suitable for the species under study. In general, only the ensemble model was compared with the boundaries of the protected areas.

Line 52: what does mean by ‘have higher quality habitat connections’?

 [Authors’ response]: Introduction: The high quality of the connection in the corridor modeling refers to the areas where there is the lowest cost of transportation. Since in this study, the inverse of the habitat suitability map is used as a cost map, so the highest quality refers to the areas with the lowest relocation costs. Therefore, because these areas have high habitat suitability, they have the lowest transportation costs. However, for ease of understanding and in accordance with the explanations mentioned in the materials and methods, “patch” was used instead of “habitat”.

Line 64: ? 4 written as superscript to ‘wild animals’

[Authors’ response]: The revision was applied. 

Line 72: ‘ in which case …’. I am unable to follow what the authors intend to say. It does need to be rephrased to make it more clarification.

[Authors’ response]: The sentence was rephrased as recommended. 

Line 76: change ‘the surest way’ to ‘the reliable way’

[Authors’ response]: The term was replaced. 

Line 83: ‘the manager’ to ‘a manager’

[Authors’ response]: The term was modified. 

Line 87: what does mean by ‘frequency’?

[Authors’ response]: frequency refers to the number of the species in the area. Modelling in the habitat suitability models can be done using either the number of a species observed in an area or only based on presence pints (regardless of number of species). The “frequency “ in this sentence was changed to “abundance”.

Line 92: explain how human pressure can contribute to the species vulnerability? 

[Authors’ response]: some content was added to the text-body explaining the impact of human pressures on species vulnerability. 

Line 95: perhaps you mean ‘reaches’ by ‘reches’? 

[Authors’ response]: the word was corrected. 

Line 96-97: needs a citation!

 [Authors’ response]: The addition of a citation here was to provide a reference to a study that specified the boundaries for the species distribution. It was to refer to a previous study as an example. 

Line 99: ‘IUCN classification’ to ‘IUCN red list’

[Authors’ response]: The revision was applied.

Lines 100-101: give a reference; however if the authors expect the negative consequences of climate changes on the species, they need to rewrite the sentence by the consideration of the possibility of incidence. 

[Authors’ response]: The sentence was modified as recommended.

Lines 103-108: the facts that have pointed out here is really important and then it needs to get involve the reliable references. 

[Authors’ response]: This part was supported by references. 

Materials and methods:

Line 115: ‘presence points’ better to change ‘species occurrences’ 

[Authors’ response]: The word was replaced.

Line 121: replace ‘symptoms’ with ‘signs’.

[Authors’ response]: The word was replaced.

Line 123: replace ‘navigating’ with ‘covering’

[Authors’ response]: The word was replaced.

Figure 1: 

- Limit the numbers of longitudinal and latitudinal coordinates around the box. 

- Steak to the instructions of the figures’ preparation guidelines of the journal to meet whatever needs to be considered to draw the figures more transparent. 

- Explain what you mean by the red crooked line and the word of ‘locality’ inset in the map?

- Generally, figure should represent more visible the main protected area, instead it has emphasised on marginal components. So, it is highly recommended to be provided more informative base map. 

 [Authors’ response]: Corrections were made. 

- “Locality” refers to the presence of species that was modified.

- Longitude was limited at the edge of the map.

- It is worth mentioning two points:

- In this study, there is not only one protected area, but also a number of areas whose boundaries are listed separately in the map. 

- The prepared map best shows the location of the protected areas in the three provinces.

Lines 142-143: rewrite as distances from gardens, main roads, streams (waterways), and human

Settlements , …

[Authors’ response]: Correction was made. 

lines 156-157: how did the authors come into that a 3×3 filter is the best spatial resolution to address the habitat fragmentation?

[Authors’ response]: The purpose of calculating this index was not to study the fragmentation of the landscape but to examine the fragmentation status of the land uses in the study area. The size of the filter depends on the input image and the purpose of the study. In this study the land use land cover maps were prepared from Landsat satellite images, which had a suitable spatial resolution (30 meters) and suitable details. Therefore, for the maximum use of these details, the mentioned filter was set to this size.

Line 161: replace ‘of’ to ‘including’ 

[Authors’ response]: the word was replaced. 

Lines 165-166: I think the 2.6 RCP is not related to the mild seanarios?!

[Authors’ response]: the authors' meant "slight”, but it was mistakenly written mild. The word has been corrected.

Lines 166-167: delete ‘Various GCM models …’ 

[Authors’ response]: the following sentence was deleted completely. 

“Various GCM models have yet been used by researchers worldwide to investigate the effect of climate change scenarios on the dispersion of plant and animal species.’

Table 2:

- The caption should reflect what the study aimed at using the environmental variables.

 [Authors’ response]: the caption was modified as recommended. 

- What was meant by ‘low’ in the third column? 

[Authors’ response]: the authors meant “range of variation”, which was mistakenly written as “low”.

- In the fourth column, what do the authors want to say by ‘Current-RCP2.6-RCP8.5’? Does not it 

 [Authors’ response]: This section tries to show which of the habitat variables have been used for what scenarios (different and current scenarios).

- That RCPs are an indication on how the green house gases’ concentration will be in the future? 

 [Authors’ response]: In each RCP, predictions were made about some climatic variables for different scenarios, which are mentioned in this table.

- So, how did you integrated those trends with the current time? 

 [Authors’ response]: there is no integration. Each scenario was run by its own variables. It is not possible to combine current data with predicted climate variables. First, the modeling was done using the current variables and then, the results of this model were generalized (projected) to future scenarios by considering the same variables.

- You can simply rescale the values of the bioclimatic variables corresponding to the temperature in the third column to make them real temperatures.

[Authors’ response]: the honourable referee is right. This is also possible through downscaling, but the honourable referee is invited to kindly pay attention to the availability as well as the limitations of meteorological stations. The low number of these stations as well as the noise in the data recording of these stations act as a barrier for a partial study (the stations in the study area were low in number and at inappropriate distances). Therefore, most researchers tend to use the data of climate database, which are based on the participation of more habitats.

Line 175: I would prefer to go with distribution instead ‘dispersion’ throughout the text. Please consider this in wherever you encounter with the dispersion in the MS. 

[Authors’ response]: the word was replaced throughout the manuscript.

Line 176: change ‘of Maxent’ to ‘including Maxent’.

[Authors’ response]: the word was replaced.

Line 176-177: it would better go for ‘To model the species distribution’ instead of ‘In the implementation of the models’

[Authors’ response]: the sentence was modified.

Line 190: Why did the authors consider only topographic heterogeneity to inspect spatially auto-correlated occurrence points despite their study is aiming at the assessment of how environmental variables (vegetation, climatic, and topographic covariates) can predict the species habitat suitability. Meanwhile, what methods did they apply to deal with this issue? For example it is strongly recommended to look at the global and local Moran coefficients (Boots, B., 2002. Local measures of spatial association. EcoScience9, 168–176).

[Authors’ response]: Most of the SDM techniques require an unbiased sample. This tool removes spatially autocorrelated occurrence points by reducing multiple occurrence records to a single record within the specified distance There are different methods to reduce the autocorrelation of presence points among the samples taken. The method used in this study considered topographic heterogeneity as an important factor in differentiating between the points of presence. It tries to keep the points that are different in terms of location based on the mentioned index and remove the other points in the radii intended by the user. The method used in this analysis is part of the SDM toolkit. The followings are further examples on the application of this methods for the same purpose.

- http://www.sdmtoolbox.org/

- Brown J.L. (2014). SDMtoolbox: a python-based GIS toolkit for landscape genetic, biogeographic, and species distribution model analyses. Methods in Ecology and Evolution.

- Hannah, L., Aguilar, G. and Blanchon, D., 2019. Spatial distribution of the Mexican daisy, Erigeron karvinskianus, in New Zealand under climate change. Climate, 7(2), p.24.

- Hannah, L., Aguilar, G. and Blanchon, D., 2019. Spatial distribution of the Mexican daisy, Erigeron karvinskianus, in New Zealand under climate change. Climate, 7(2), p.24.

Line 204-205: How was the ensemble map created?

[Authors’ response]: the map was prepared in ModEco software in two steps as depicted below: 

Figure 2: Different steps of using the Ensemble model

After the mentioned models were converted to binary maps based on the cut-off value, they were combined using all the maps, and from the path of Figure 1. By choosing the method of “accuracy rate as weight” based on the AUC index the integration was done in two steps of 1 and 2.

Therefore, in this study, the ensemble model has been formed twice:

1- For presence-only methods (Figure 1) in order to achieve pseudo-absence points

2- For modeling using presence / pseudo-absence points (Figure 2).

Line 207-208: I am not sure why the authors intend to relate the species adaptation to climate change to the current protected areas (PAs) network. Are they sure the current PAs will sufficiently meet the future species climate requirements?. Specifically, the authors need to explicitly outline the biological justifications that they aim to address through this investigation. 

[Authors’ response]: The purpose of these sentences is to determine the efficiency of the current boundary range of protected areas compared to the conditions that may occur in climate change scenarios. In other words, the main purpose of referring to this issue is to determine the efficiency of the boundary of protected areas for possible scenarios. Therefore, in the continuation of the article, in Table 5, the amount of coverage of the habitats by the border of protected areas is specified. Therefore, the main purpose is to evaluate the efficiency of the current network of protected areas in climate change scenarios. A similar analysis has been made in most climate change predictions. For example the following study:

- Karami P, Rezaei S, Shadloo S, Naderi M. An evaluation of central Iran’s protected areas under different climate change scenarios (A Case on Markazi and Hamedan provinces). Journal of Mountain Science 2020;17(1): 68-82.

- Ahmadi, M., Nezami Balouchi, B., Jowkar, H., Hemami, M.R., Fadakar, D., Malakouti‐Khah, S. and Ostrowski, S., 2017. Combining landscape suitability and habitat connectivity to conserve the last surviving population of cheetah in Asia. Diversity and Distributions, 23(6), pp.592-603.

- Su, J., Aryal, A., Hegab, I.M., Shrestha, U.B., Coogan, S.C., Sathyakumar, S., Dalannast, M., Dou, Z., Suo, Y., Dabu, X. and Fu, H., 2018. Decreasing brown bear (Ursus arctos) habitat due to climate change in Central Asia and the Asian Highlands. Ecology and Evolution, 8(23), pp.11887-11899.

- Kafaei, S., Akmali, V. and Sharifi, M., 2020. Using the Ensemble Modeling Approach to Predict the Potential Distribution of the Muscat Mouse-Tailed Bat, Rhinopoma muscatellum (Chiroptera: Rhinopomatidae), in Iran. Iranian Journal of Science and Technology, Transactions A: Science, pp.1-12.

- Sharifi, M., Karami, P., Akmali, V., Afroosheh, M. and Vaissi, S., 2017. Modeling geographic distribution for the endangered yellow spotted mountain newt, Neurergus microspilotus (Amphibia: Salamandridae) in iran and Iraq. Herpetological Conservation and Biology, 12(2), pp.488-497.

In accordance with the opinion of the esteemed referee, the objectives of the article were modified in the introduction section.

Line 214: change ‘circuitscape’ to ‘Circuitscape’

[Authors’ response]: it was corrected. 

Line 214-218: This section would benefit from some more references to back up and provide further detail. 

[Authors’ response]: more references were added to support this section. 

Line 219: what is ‘The cost or Friction map’? would be added a citation! In addition, the authors sharply jumped into this paragraph without a logical transition from the later paragraph. It would be better rephrase these sections to follow the authors’ thoughts more easily. 

[Authors’ response]: More details about cost or friction map was added. Attempts were made to include the mentioned explanations in the text in an orderly and consistent manner in terms of meaning.

Line 221: What is a difference between the cost and resistance maps?

[Authors’ response]: They have no difference, and both point to the cost that the species must pay in the transmission path. 

Line 225: Add reference(s) to ‘or as the boundary of protected areas.’

[Authors’ response]: The revision was applied. This section was revised. Focal nodes in the software menu is a section that introduces habitat patches or connecting cores. In this study, protected areas were considered as focal nodes.

Line 225: What does mean by ‘focal nodes’?

 [Authors’ response]: The revision was applied. This part was revised due to incorrect translation. Focal nodes in the software menu is the part where the user introduces habitat spots or connection cores, which in this study, protected areas are considered as Focal nodes.

Lines 212-229: The use of ‘electronic circuit theory’ in the Methods needs more development in terms of recasting in a way that can be interpreted ecologically. Currently it reads very much as having applied a "black box" with foreign (to wildlife research at least) terms such as "current flow" and "resistance". Can these be recast as more accessible descriptions for the reader?

[Authors’ response]: The electrical circuit theory section was revised.

Line 231: I am really confused why the authors did employ a modeling technique (random forest) to see the contribution importance of the environmental variables in the species occurrence probability that was not included in an array of the applied modeling methods i.e. GLM, Maxent, SVM, and BP-ANN (lines 175-176) which were used to predict the species distribution.

[Authors’ response]: The habitat modeling software used in this study has several advantages, including having multiple modeling methods. In fact, this software offers many models used in the habitat modeling process. The ease of use and user-friendly graphical environment have led to easy use of this software. The software used in the model validation and threshold identification process works well, but there is no option to check the importance of the variables in its models. 

 

As shown in Figure 3, to determine the importance of different habitat variables, this software can only be implemented using a specific set of models.

Figure 3: Constraint of ModEco software in determining the importance of habitat variables

It was not possible to determine the importance of habitat variables in accordance with the models used in the modeling process using this software. Therefore, it was decided to use an alternative method that can appropriately determine the effect of habitat variables in each model. There are different methods in this field, but since random forest method is one of the widely-used methods (Cutler et al.,2007; Strobl et al.,2007) in identifying the importance of variables, it was used to investigate the importance of variables in each model.

- Cutler, D.R., Edwards Jr, T.C., Beard, K.H., Cutler, A., Hess, K.T., Gibson, J. and Lawler, J.J., 2007. Random forests for classification in ecology. Ecology, 88(11), pp.2783-2792.

- Strobl, C., Boulesteix, A.L., Zeileis, A. and Hothorn, T., 2007. Bias in random forest variable importance measures: Illustrations, sources and a solution. BMC bioinformatics, 8(1), p.25.

Lines 232-242: This section can be shortened more logically by adding more relevant refrence(s) to facilitate reader to follow the authors’ thoughts.

 [Authors’ response]: this section was revised as recommended. 

Line 241-242: This is extremely good to see the authors are using more applicable scientific program (R) here, by considering this point, I am really baffling why they did not use the potential of R program which is publically free available through more compressive species distribution analyzing packages to take into account the latest developments in species distribution modeling. I truly believe that this study will have meaningful contributions to the species ecology and conservation if the authors would polish their analyses in R!. 

[Authors’ response]: The words of the honourable referee are absolutely correct. However, there are reasons why authors tend to use MODECO software.

 1. Preparing pseudo-absence points and variety of existing methods:

Creating pseudo-absence points seamlessly is only possible in this software. This software provides the possibility to integrate different presence-only methods in order to achieve pseudo-absence points, which has not been included in various habitat modeling packages in the R environment.

2. Multiplicity of available 

In this software, more models are available for use than the packages in R software. In fact, almost all habitat modeling methods are included in this software.

3. Being fast and availability of different models:

In this software, the available models can be used freely and run very quickly.

4. Possibility to determine the type of forecast

In this software, it is possible for the user to determine the type of prediction as probable (probability of presence) or definite (zero and one). The following figure shows the available models for use.

Figure 4: models available in ModEco

Line 247 and Table 3: It is very interesting that I see the authors are using the significant metric (p-value) to address their findings. But it would be really useful to be added more descriptions in the methodology part of the MS how they did statistically compare the models performances using the evaluation metrics. This analysis would be more reproducible if the authors back their analysis given a reference(s). It is urgently needed to add more clarification on it since this is very important point. 

 [Authors’ response]: This part of the analysis is related to the norm hypothesis used to calculate the AUC, which was mentioned in the results section at the request of the Referee. 

Line 256: ‘the different RCP 2.6 scenarios’: It is not more clear. How many scenarios are available for RCP 2.6.? I assume that there is just one RCP 2.6! 

[Authors’ response]: This study considered three scenarios, including CCSM4, BCC-CSM1-1, and MRI-CGCM3 to predict future changes, each of which run under RCP2.6 and RCP8.5. So, the there are 3 RCP2.6 and 3 RCP8.5 in this study (Figure 3).

Line 258: ‘Under the RCP8.5 scenario, the habitat suitability declines in the northern parts…’. How do the authors come to this finding? It is worthy to be specific here by giving more information where and how many suitable patches will be lost under this scenario? AS the ms stands (lines 258-260), it is not scientifically informative to visually interpret the change trends of the species habitat suitability. 

[Authors’ response]: these are the presence probability maps with values ranging from 0 to 1 on a probability basis, so they cannot be analyzed quantitatively. This is why thresholds are applied to habitat suitability maps to quantify results. The complete and complementary results of this section are listed in Table 5.

Lines 265-266: How did the authors go straight to this point? Did they know where the suitable/unsuitable areas occur already to draw this interpretation?

[Authors’ response]: After identifying the cut-off value and applying it to the habitat suitability map, the map of suitable/unsuitable habitats were obtained. Then, the suitability values (1 and 0) of the habitat suitability map for the presence / pseudo-absence points were extracted. Accordingly, the metrics in Table 4 were easily calculated.

Table 4: What does mean by greater than for the thresholds of the model in the second column? Did the authors use these threshold levels to binaries the continuous suitability maps? If so, how? And also for what metrics (TSS, AUC, Kappa, …) was these levels calculated? It is really important to be awareness on uncertainties inherits in developing the models. So, the authors are strangely recommended to iterate each model several times (e.g. 30 times) to ensure that the results are expressed statistically meaningful in terms of the confidence intervals when it goes to consider the model performances using the elevation metrics. 

Answers to each questions:

1- What does mean by greater than for the thresholds of the model in the second column? Did the authors use these threshold levels to binaries the continuous suitability maps? If so, how?

[Authors’ response]: These threshold values are detected by SPSS software; and are usually identified as such. For example,> 0.26 shows the threshold at which 92% of the presence points are identified as presence points and 82% of the pseudo-non-presence points are identified as pseudo-non-presence points. This value was calculated based on the TSS threshold formula as follows (Allouche ET AL., 2006).

TSS= sensitivity + specificity – 1

Allouche, O., Tsoar, A. and Kadmon, R., 2006. Assessing the accuracy of species distribution models: prevalence, kappa and the true skill statistic (TSS). Journal of applied ecology, 43(6), pp.1223-1232.

This can be done using the raster calculator in the ArcMap software.

2- And also for what metrics (TSS, AUC, Kappa, …) was these levels calculated?

[Authors’ response]: Some of the mentioned metrics are threshold dependent and some are non-threshold dependent, which should be distinguished between the two groups of metrics. The AUC value is calculated using the ROC curve. The TSS statistic has two values, one related to validation and the other related to the threshold. Its numerical value in Table 3 can be interpreted with the explanations in the materials and methods section. Another value that can be calculated based on the TSS formula is the threshold, the exact value of which for each model run is listed in Table 4. After identifying the numerical value of the threshold limit, this value was applied on the suitability maps and the value of the resulting map (zero and one map obtained from the threshold) are obtained with the values of 0 and 1 of the pseudo-absence points and the values of the relevant metrics.

3- It is really important to be awareness on uncertainties inherits in developing the models. So, the authors are strangely recommended to iterate each model several times (e.g. 30 times) to ensure that the results are expressed statistically meaningful in terms of the confidence intervals when it goes to consider the model performances using the elevation metrics. 

[Authors’ response]: One of the reasons that Modeco software was chosen for modeling, as mentioned above, was its repeatability. In fact, the authors run these models for several times to get the best results. So, the valuable recommendation of the honourable Referee has already been applied when running the model.

Lines 277 and 278: What do the authors mean by the similarities? On base of what quantity metric(s) do they go for through the more similarities’ claim? 

[Authors’ response]: The mean was similarity between suitable and unsuitable habitats. This part was excluded from the results. 

-The terms ‘current flow’ (line 227) and ‘amps’ (line 299) seem a misnomer. Can you express in a more ecological manner that makes more sense to the reader? For example, the term ‘connectivity’ has an ecological meaning, but ‘current flow’ doesn’t at this stage.

 [Authors’ response]: In this section, an attempt was made to explain the results based on the ecological concepts and terms as recommended.

Line 323: What do bio19, 17, and 7 stand for? Here it is good to define these abbreviations. How do species response to these climatic variables. Creating response curves for them would add more value to the study.

[Authors’ response]: Table 2, which deals with habitat variables used, has been revised and the full name of the variables was added to the table. In order to prevent the increase of figures in the article, these figures are presented only in the Rebuttal letter not in the manuscript. The following figure shows the response curve of the 3 main influential variables.

Figure 4: Response curves of the 3 main variables affecting the species presence

Line 327: Capitalize ‘random’ 

[Authors’ response]: it was capitalized. 

-table 6: Remove unnecessary horizontal and vertical lines. What is ‘Peasant’? maybe you mean ‘present’. Uncertainty on these findings could be controlled if the authors performed multi replicates (e.g. 30 times) of their model as I mentioned before in developing species distribution modeling (please also note the comments on Table 4). 

 [Authors’ response]: the table was modified as recommended. One of the advantages of the software used in this study is replicability. The models mentioned in this study are the best models that are known to be appropriate in replication.

# General comments on figures: The figures are predominantly drawn in a way making it hard to interpret the main points what the authors would intend to present. Generally, what is highlighted in each figure exactly is not the main goal of the study but some main points that are needed to be more visible and clearer; unfortunately, they did not take into consideration for them. 

 [Authors’ response]: the figures of the paper have been revised as recommended.

---

## [Editor Report · Decision Letter 2]

27 Oct 2020

PONE-D-20-07352R2

Accessing Habitat Suitability and Connectivity for the Westernmost Population of Asian Black Bear (Ursus thibetanus gedrosianus, Blanford, 1877) based on Climate Changes Scenarios in Iran

PLOS ONE

Dear Dr. Morovati,

Thank you for submitting your manuscript to PLOS ONE. After careful consideration, we feel that it has merit but does not fully meet PLOS ONE’s publication criteria as it currently stands. Therefore, we invite you to submit a revised version of the manuscript that addresses the points raised during the review process.

We look forward to receiving your revised manuscript.

Kind regards,

Lyi Mingyang, Ph.D.

Academic Editor

PLOS ONE

Additional Editor Comments (if provided):

Dear Authors,

I noticed that you have replied only to Reviewer2 completely ignoring what was requested by the Reviewer 1.

I will ask to you to also address all the few comments made by the Reviewer 1 (please, see below).

It is a point of respect and education of the time spent by the reviewer 1 on this manuscipt.

Reviewer1

General comments:

The manuscript “Accessing Habitat Suitability and Connectivity for the Easternmost Population of Asian Black Bear (Ursus thibetanus gedrosianus, Blanford, 1877) based on Climate Changes Scenarios in Iran”, submitted to PLOSONE by Morovati et al., has definitely improved after the first revision, however, the authors have yet to work long before that this paper can be definitively accepted. This ms provides interesting information but it needs to be improved. I think that some references should be added in some specific point of the text. Moreover, this manuscript should be absolutely revise by a English Native Speaker. Please, see below my specific comments.

Specific comments:

Line 30: To use the acronyms (e.g., GLM) only if you will use them another time in the abstract. Please, check the acronyms in all the abstract.

Line 64: [4] not superscript.

Lines 81 – 82: I think that you should add some references as examples to support this your sentence “Models with the ability to predict the suitability of wildlife habitats on a large scale can be useful for wildlife managers.” I would like to suggest:

Bertolino S., et al., (2020). Spatially-explicit models as tools for implementing effective management strategies for invasive alien mammals. Mammal Review, 50, 187-199.

Pauli, B. P., et al. (2019). Human habitat selection: using tools from wildlife ecology to predict recreation in natural landscapes. Natural Areas Journal, 39(2), 142-149.

Lines 83 – 84: I think that you should add some references as examples to support this your sentence “For the protection of an important species, it is of critical importance to identify its needs as well as habitat constraints, and degradation factors.” I would like to suggest:

Smeraldo, S., et al., (2020). Modelling risks posed by wind turbines and power lines to soaring birds: the black stork (Ciconia nigra) in Italy as a case study. Biodiversity and Conservation, 29, 1959-1976.

Andrade-Díaz, M. S., et al., (2019). Expansion of the agricultural frontier in the largest South American Dry Forest: Identifying priority conservation areas for snakes before everything is lost. PloS one, 14(9), e0221901.

Lines 182 – 183: I think that you should add some references as examples to support this your sentence “After identifying the threshold, the metrics of sensitivity, specificity, correct classification, and miss classification were used to assess the power of the threshold [3].” I would like to suggest:

Ancillotto, L., et al. (2020). An African bat in Europe, Plecotus gaisleri: Biogeographic and ecological insights from molecular taxonomy and Species Distribution Models. Ecology and Evolution, 10, 5785-5800.

Lines 209 – 210: This figure should be moved in the Results. To delete 0’0” from the figure.

Line 327: Why did you colour some number in grey?

Best regards,

LM

---

## [Author Response · Author response to Decision Letter 2]

30 Oct 2020

Ms.No.PONE-D-20-07352

There were several parts of the manuscript (MS) that were confusing and unfocused despite reviewer#2 highlighted several major issues, especially those pertaining to methodological and modeling concerns including the delimitation of ‘model calibration region’ and ‘Model uncertainty’. As the MS stands, much of the confusion seems to come from methods and results sections yet because the whole paper did not comprehensively follow reviewer’s comments. Some of my specific comments below may help, but in general, the authors should try to improve readability and clarity by thinking about the organization of the paper and the logical transitional flow between ideas.

Dear Editor,

We hereby appreciate valuables comments. Before answering getting started, let me provide you with some explanations about the first-round revisions. In the first round of revising the paper, objections and suggestions were made by the first and second Referees, the response to which required a major and complete revision of the article. According to the first-round revisions, the concerns of both Referees have been answered and applied in the paper, which generally included the following items:

1- All sections of the article such as abstract, introduction, materials and methods, results and discussion, and conclusion were revised completely.

2- Instead of using one model, several models have been applied.

3- Instead of using one climate change scenario, several different models under the RCP2.6 and RCP8.5 scenarios have been used.

4- The efficiency of the network of the protected areas in covering the suitable habitats has also been studied. 

The changes will be obvious if this version is compared to the previous one, as we invite to kindly do so.

Thank you again for the time spending to read the rebuttal letter.

Hope the revisions would be satisfactory enough to meet the satisfaction of honorable Editor and Referees.

Truly yours

The authors

Reviewer1

General comments:

The manuscript “Accessing Habitat Suitability and Connectivity for the Easternmost Population of Asian Black Bear (Ursus thibetanus gedrosianus, Blanford, 1877) based on Climate Changes Scenarios in Iran”, submitted to PLOSONE by Morovati et al., has definitely improved after the first revision, however, the authors have yet to work long before that this paper can be definitively accepted. This ms provides interesting information but it needs to be improved. I think that some references should be added in some specific point of the text. Moreover, this manuscript should be absolutely revise by a English Native Speaker. Please, see below my specific comments.

Greetings and respect

Dear Judge, The authors of the article are grateful for the time you spent reading this work and your valuable suggestion. The items of the dear judge are marked in yellow on the body of the article. The following are your comments and the authors' answers

.

Specific comments:

Line 30: To use the acronyms (e.g., GLM) only if you will use them another time in the abstract. Please, check the acronyms in all the abstract.

[Authors’ response]: Abbreviations in the abstract were examined.

.

Line 64: [4] not superscript.

[Authors’ response]: the revision was applied.

Lines 81 – 82: I think that you should add some references as examples to support this your sentence “Models with the ability to predict the suitability of wildlife habitats on a large scale can be useful for wildlife managers.” I would like to suggest:

Bertolino S., et al., (2020). Spatially-explicit models as tools for implementing effective management strategies for invasive alien mammals. Mammal Review, 50, 187-199.

Pauli, B. P., et al. (2019). Human habitat selection: using tools from wildlife ecology to predict recreation in natural landscapes. Natural Areas Journal, 39(2), 142-149.

[Authors’ response]: Suggested references were added to the text.

Lines 83 – 84: I think that you should add some references as examples to support this your sentence “For the protection of an important species, it is of critical importance to identify its needs as well as habitat constraints, and degradation factors.” I would like to suggest:

Smeraldo, S., et al., (2020). Modelling risks posed by wind turbines and power lines to soaring birds: the black stork (Ciconia nigra) in Italy as a case study. Biodiversity and Conservation, 29, 1959-1976.

Andrade-Díaz, M. S., et al., (2019). Expansion of the agricultural frontier in the largest South American Dry Forest: Identifying priority conservation areas for snakes before everything is lost. PloS one, 14(9), e0221901.

[Authors’ response]: Suggested references were added to the text.

Lines 182 – 183: I think that you should add some references as examples to support this your sentence “After identifying the threshold, the metrics of sensitivity, specificity, correct classification, and miss classification were used to assess the power of the threshold [3].” I would like to suggest:

Ancillotto, L., et al. (2020). An African bat in Europe, Plecotus gaisleri: Biogeographic and ecological insights from molecular taxonomy and Species Distribution Models. Ecology and Evolution, 10, 5785-5800.

[Authors’ response]: Suggested references were added to the text.

Lines 209 – 210: This figure should be moved in the Results. To delete 0’0” from the figure.

[Authors’ response]: This figure shows the pseudo-absence points that are made, which is a prerequisite for running MaxEnt, GLM, SVM and BP-ANN models. Therefore, it is part of the materials and methods of this study. Hence it is mentioned in the materials and methods section.

Line 327: Why did you colour some number in grey?

[Authors’ response]: The color of some cells of the table, define the variables that have the greatest impact on the Random Forest method according to the presence or absence of the species.

Reviewer2

Greetings and respect

Dear Judge, The authors of the article are grateful for the time you spent reading this work and your valuable suggestion. The items of the dear judge are marked in green on the body of the article. The following are your comments and the authors' answers

SPECIFIC/MINOR ISSUES

Line 25: change ‘communications’ with ‘connectivity’

[Authors’ response]: the revision was applied.

Line 28: remove ‘For this purpose’ 

[Authors’ response]: the revision was applied.

Line 28: change ‘3 provinces’ to ‘provinces’

[Authors’ response]: the revision was applied.

Line 29: change ‘Sistan and Baluchestan’ to ‘Sistan-Baluchestan’. Follow this afterwards throughout the text.

[Authors’ response]: the revision was applied.

Line 29: remove ‘and then, corrected in terms of spatial autocorrelation’

[Authors’ response]: the revision was applied.

Lines 30-31: delete the abbreviations of modelling techniques

[Authors’ response]: the revision was applied.

Line 30: change ‘,3 machine learning models of’ to ‘ and 3 machine learning models including’

[Authors’ response]: the revision was applied.

Line 31: Ensemble model is not itself a unique modelling algorithm, it is just an approach to combine the results of formal habitat modelling algorithms. So, the authors should address how the ensemble model was undertaken to integrate the main habitat models as a separate sentence. 

[Authors’ response]: thank you very much for the comment. We used the term of “model” is referring for the following reasons:

Ensemble in ModEco is introduced as a model with a meta-algorithm:

“- Bootstrap aggregating (bagging) is a machine learning ensemble meta-algorithm to improve classification and regression models in terms of stability and classification accuracy. 

- AdaBoost, short for Adaptive Boosting, is a meta-algorithm, and can be used in conjunction with many other learning algorithms to improve their performance.” 

In general, depending on the type of model used, the ensemble can be a very simple algorithm or use mixed algorithms in its background for modeling. Ensemble is referred as “Model” in the software interface. Please take a look to the following print screen:

Figure 1: Ensemble model direction in ModEco software

There are many articles referring to Ensemble as a model:

Guo, Q. and Liu, Y., 2010. ModEco: an integrated software package for ecological niche modeling. Ecography, 33(4), pp.637-642.

- Kafaei, S., Akmali, V. and Sharifi, M., 2020. Using the Ensemble Modeling Approach to Predict the Potential Distribution of the Muscat Mouse-Tailed Bat, Rhinopoma muscatellum (Chiroptera: Rhinopomatidae), in Iran. Iranian Journal of Science and Technology, Transactions A: Science, pp.1-12.

- Lei, J., Chen, L. and Li, H., 2017. Using ensemble forecasting to examine how climate change promotes worldwide invasion of the golden apple snail (Pomacea canaliculata). Environmental Monitoring and Assessment, 189(8), p.404.

- Mugo, R. and Saitoh, S.I., 2020. Ensemble Modelling of Skipjack Tuna (Katsuwonus pelamis) Habitats in the Western North Pacific Using Satellite Remotely Sensed Data; a Comparative Analysis Using Machine-Learning Models. Remote Sensing, 12(16), p.2591.

Anyhow, according to the Referee’s comment, we modified the “Ensemble model” in the paper as Ensemble modeling approach.

Lines 32-34: I am not sure that I could understand well what the authors mean by how they generated the background points to undertake the habitat modelling. It would better not mention it here and go through to explain it in depth in the appropriate part of the MS. 

[Authors’ response]: In this study, pseudo-absence points were prepared based on the Ensemble approach and presence-only models, i.e. the presence-only models were considered, and then modeling was done using these models. Finally, the results of these models were integrated. The map obtained from the combination of these models was used to prepare pseudo-absence points. 

According to the referee’s comment, the sentence explaining this process was removed and replaced by a short sentence.

Line 36: remove ‘,which have the best response for Iran,’

[Authors’ response]: the sentence was deleted. 

Lines 39-40: rephrase as: ‘The true skill statistic was employed to convert the continuous suitability layers to binary suitable/unsuitable range maps to assess the effectiveness of the protected areas in the coverage of suitable habitats for the species.’ It should point out why the authors chose this evaluation metric to dichotomize the habitat suitability, not here but the next parts of the MS.

[Authors’ response]: the abstract was modified and the explanation was added to the MS where appropriate. 

Lines 40-41: better to rephrase as: ‘The random forest method was also used to predict the relationships between the presence probability of the species and the environmental predicators.’ But, it should be highlighted in the other sections of the MS not here the reasons behind selecting this model to evaluate the species response curves to the predictors. 

 [Authors’ response]: the abstract was modified as recommended and in the text-body, it was explained that this model was used due to its high efficiency in identifying the influential variables. 

Line 42: To be specific. which model(s)? Did all models perform well in discriminating suitable and unsuitable areas?

 [Authors’ response]: the mean was the presence/pseudo-absence models used in the modeling process as mentioned in the abstract. 

Line 42: what does mean by ‘successful in the implementation’?

 [Authors’ response]: the mean by "Successful in the implementation" was “the successful run of the used models as they showed acceptable values of sensitivity and specificity. A similar term is found in other examples such as the followings:

- Cao, Y., DeWalt, R.E., Robinson, J.L., Tweddale, T., Hinz, L. and Pessino, M., 2013. Using Maxent to model the historic distributions of stonefly species in Illinois streams: the effects of regularization and threshold selections. Ecological Modelling, 259, pp.30-39.

- Fukuda, S., De Baets, B., Mouton, A.M., Waegeman, W., Nakajima, J., Mukai, T., Hiramatsu, K. and Onikura, N., 2011. Effect of model formulation on the optimization of a genetic Takagi–Sugeno fuzzy system for fish habitat suitability evaluation. Ecological modelling, 222(8), pp.1401-1413.

Line 43: be specific in what bio17, … stands for? Then delete the words of bio17, … here.

 [Authors’ response]: the complete name of all variables was included in the abstract.

Lines 43-44: clarify what the authors mean by ‘high’ and ‘extreme’? 

 [Authors’ response]: This sentence is based on the results of modeling and distribution of suitable habitats of the species in the study area, which is also mentioned in the discussion and conclusion. However, this sentence was corrected in the abstract.

Line 45: To which habitat modelling techniques were the area of suitable habitats shown in parentheses associated?

 [Authors’ response]: The area calculation was done only based upon the Ensemble models because Ensemble was used to generalize (prediction) the modeling results.

Line 46: replace ‘higher’ with ‘larger’ and ‘the current’ with ‘the current distribution’.

 [Authors’ response]: The revisions were applied as recommended. 

Line 46: ‘all the models run under…’. I am completely confused about why the authors applied the ensemble model regardless they are interested in addressing the changes of the species habitat suitability through the application of every single model? by considering that the main idea behind using the ensemble modelling is to deal with the uncertainties inherited from each modelling algorithm. 

[Authors’ response]: In this sentence, the authors meant that in each model a decrease was observed in the full coverage of suitable habitats of the species by the protected areas. This sentence was presented here to say that in all of the models, the network of the protected areas failed to fully cover the habitats suitable for the species under study. In general, only the ensemble model was compared with the boundaries of the protected areas.

Line 52: what does mean by ‘have higher quality habitat connections’?

 [Authors’ response]: Introduction: The high quality of the connection in the corridor modeling refers to the areas where there is the lowest cost of transportation. Since in this study, the inverse of the habitat suitability map is used as a cost map, so the highest quality refers to the areas with the lowest relocation costs. Therefore, because these areas have high habitat suitability, they have the lowest transportation costs. However, for ease of understanding and in accordance with the explanations mentioned in the materials and methods, “patch” was used instead of “habitat”.

Line 64: ? 4 written as superscript to ‘wild animals’

[Authors’ response]: The revision was applied. 

Line 72: ‘ in which case …’. I am unable to follow what the authors intend to say. It does need to be rephrased to make it more clarification.

[Authors’ response]: The sentence was rephrased as recommended. 

Line 76: change ‘the surest way’ to ‘the reliable way’

[Authors’ response]: The term was replaced. 

Line 83: ‘the manager’ to ‘a manager’

[Authors’ response]: The term was modified. 

Line 87: what does mean by ‘frequency’?

[Authors’ response]: frequency refers to the number of the species in the area. Modelling in the habitat suitability models can be done using either the number of a species observed in an area or only based on presence pints (regardless of number of species). The “frequency “ in this sentence was changed to “abundance”.

Line 92: explain how human pressure can contribute to the species vulnerability? 

[Authors’ response]: some content was added to the text-body explaining the impact of human pressures on species vulnerability. 

Line 95: perhaps you mean ‘reaches’ by ‘reches’? 

[Authors’ response]: the word was corrected. 

Line 96-97: needs a citation!

 [Authors’ response]: The addition of a citation here was to provide a reference to a study that specified the boundaries for the species distribution. It was to refer to a previous study as an example. 

Line 99: ‘IUCN classification’ to ‘IUCN red list’

[Authors’ response]: The revision was applied.

Lines 100-101: give a reference; however if the authors expect the negative consequences of climate changes on the species, they need to rewrite the sentence by the consideration of the possibility of incidence. 

[Authors’ response]: The sentence was modified as recommended.

Lines 103-108: the facts that have pointed out here is really important and then it needs to get involve the reliable references. 

[Authors’ response]: This part was supported by references. 

Materials and methods:

Line 115: ‘presence points’ better to change ‘species occurrences’ 

[Authors’ response]: The word was replaced.

Line 121: replace ‘symptoms’ with ‘signs’.

[Authors’ response]: The word was replaced.

Line 123: replace ‘navigating’ with ‘covering’

[Authors’ response]: The word was replaced.

Figure 1: 

- Limit the numbers of longitudinal and latitudinal coordinates around the box. 

- Steak to the instructions of the figures’ preparation guidelines of the journal to meet whatever needs to be considered to draw the figures more transparent. 

- Explain what you mean by the red crooked line and the word of ‘locality’ inset in the map?

- Generally, figure should represent more visible the main protected area, instead it has emphasised on marginal components. So, it is highly recommended to be provided more informative base map. 

 [Authors’ response]: Corrections were made. 

- “Locality” refers to the presence of species that was modified.

- Longitude was limited at the edge of the map.

- It is worth mentioning two points:

- In this study, there is not only one protected area, but also a number of areas whose boundaries are listed separately in the map. 

- The prepared map best shows the location of the protected areas in the three provinces.

Lines 142-143: rewrite as distances from gardens, main roads, streams (waterways), and human

Settlements , …

[Authors’ response]: Correction was made. 

lines 156-157: how did the authors come into that a 3×3 filter is the best spatial resolution to address the habitat fragmentation?

[Authors’ response]: The purpose of calculating this index was not to study the fragmentation of the landscape but to examine the fragmentation status of the land uses in the study area. The size of the filter depends on the input image and the purpose of the study. In this study the land use land cover maps were prepared from Landsat satellite images, which had a suitable spatial resolution (30 meters) and suitable details. Therefore, for the maximum use of these details, the mentioned filter was set to this size.

Line 161: replace ‘of’ to ‘including’ 

[Authors’ response]: the word was replaced. 

Lines 165-166: I think the 2.6 RCP is not related to the mild seanarios?!

[Authors’ response]: the authors' meant "slight”, but it was mistakenly written mild. The word has been corrected.

Lines 166-167: delete ‘Various GCM models …’ 

[Authors’ response]: the following sentence was deleted completely. 

“Various GCM models have yet been used by researchers worldwide to investigate the effect of climate change scenarios on the dispersion of plant and animal species.’

Table 2:

- The caption should reflect what the study aimed at using the environmental variables.

 [Authors’ response]: the caption was modified as recommended. 

- What was meant by ‘low’ in the third column? 

[Authors’ response]: the authors meant “range of variation”, which was mistakenly written as “low”.

- In the fourth column, what do the authors want to say by ‘Current-RCP2.6-RCP8.5’? Does not it 

 [Authors’ response]: This section tries to show which of the habitat variables have been used for what scenarios (different and current scenarios).

- That RCPs are an indication on how the green house gases’ concentration will be in the future? 

 [Authors’ response]: In each RCP, predictions were made about some climatic variables for different scenarios, which are mentioned in this table.

- So, how did you integrated those trends with the current time? 

 [Authors’ response]: there is no integration. Each scenario was run by its own variables. It is not possible to combine current data with predicted climate variables. First, the modeling was done using the current variables and then, the results of this model were generalized (projected) to future scenarios by considering the same variables.

- You can simply rescale the values of the bioclimatic variables corresponding to the temperature in the third column to make them real temperatures.

[Authors’ response]: the honourable referee is right. This is also possible through downscaling, but the honourable referee is invited to kindly pay attention to the availability as well as the limitations of meteorological stations. The low number of these stations as well as the noise in the data recording of these stations act as a barrier for a partial study (the stations in the study area were low in number and at inappropriate distances). Therefore, most researchers tend to use the data of climate database, which are based on the participation of more habitats.

Line 175: I would prefer to go with distribution instead ‘dispersion’ throughout the text. Please consider this in wherever you encounter with the dispersion in the MS. 

[Authors’ response]: the word was replaced throughout the manuscript.

Line 176: change ‘of Maxent’ to ‘including Maxent’.

[Authors’ response]: the word was replaced.

Line 176-177: it would better go for ‘To model the species distribution’ instead of ‘In the implementation of the models’

[Authors’ response]: the sentence was modified.

Line 190: Why did the authors consider only topographic heterogeneity to inspect spatially auto-correlated occurrence points despite their study is aiming at the assessment of how environmental variables (vegetation, climatic, and topographic covariates) can predict the species habitat suitability. Meanwhile, what methods did they apply to deal with this issue? For example it is strongly recommended to look at the global and local Moran coefficients (Boots, B., 2002. Local measures of spatial association. EcoScience9, 168–176).

[Authors’ response]: Most of the SDM techniques require an unbiased sample. This tool removes spatially autocorrelated occurrence points by reducing multiple occurrence records to a single record within the specified distance There are different methods to reduce the autocorrelation of presence points among the samples taken. The method used in this study considered topographic heterogeneity as an important factor in differentiating between the points of presence. It tries to keep the points that are different in terms of location based on the mentioned index and remove the other points in the radii intended by the user. The method used in this analysis is part of the SDM toolkit. The followings are further examples on the application of this methods for the same purpose.

- http://www.sdmtoolbox.org/

- Brown J.L. (2014). SDMtoolbox: a python-based GIS toolkit for landscape genetic, biogeographic, and species distribution model analyses. Methods in Ecology and Evolution.

- Hannah, L., Aguilar, G. and Blanchon, D., 2019. Spatial distribution of the Mexican daisy, Erigeron karvinskianus, in New Zealand under climate change. Climate, 7(2), p.24.

- Hannah, L., Aguilar, G. and Blanchon, D., 2019. Spatial distribution of the Mexican daisy, Erigeron karvinskianus, in New Zealand under climate change. Climate, 7(2), p.24.

Line 204-205: How was the ensemble map created?

[Authors’ response]: the map was prepared in ModEco software in two steps as depicted below: 

Figure 2: Different steps of using the Ensemble model

After the mentioned models were converted to binary maps based on the cut-off value, they were combined using all the maps, and from the path of Figure 1. By choosing the method of “accuracy rate as weight” based on the AUC index the integration was done in two steps of 1 and 2.

Therefore, in this study, the ensemble model has been formed twice:

1- For presence-only methods (Figure 1) in order to achieve pseudo-absence points

2- For modeling using presence / pseudo-absence points (Figure 2).

Line 207-208: I am not sure why the authors intend to relate the species adaptation to climate change to the current protected areas (PAs) network. Are they sure the current PAs will sufficiently meet the future species climate requirements?. Specifically, the authors need to explicitly outline the biological justifications that they aim to address through this investigation. 

[Authors’ response]: The purpose of these sentences is to determine the efficiency of the current boundary range of protected areas compared to the conditions that may occur in climate change scenarios. In other words, the main purpose of referring to this issue is to determine the efficiency of the boundary of protected areas for possible scenarios. Therefore, in the continuation of the article, in Table 5, the amount of coverage of the habitats by the border of protected areas is specified. Therefore, the main purpose is to evaluate the efficiency of the current network of protected areas in climate change scenarios. A similar analysis has been made in most climate change predictions. For example the following study:

- Karami P, Rezaei S, Shadloo S, Naderi M. An evaluation of central Iran’s protected areas under different climate change scenarios (A Case on Markazi and Hamedan provinces). Journal of Mountain Science 2020;17(1): 68-82.

- Ahmadi, M., Nezami Balouchi, B., Jowkar, H., Hemami, M.R., Fadakar, D., Malakouti‐Khah, S. and Ostrowski, S., 2017. Combining landscape suitability and habitat connectivity to conserve the last surviving population of cheetah in Asia. Diversity and Distributions, 23(6), pp.592-603.

- Su, J., Aryal, A., Hegab, I.M., Shrestha, U.B., Coogan, S.C., Sathyakumar, S., Dalannast, M., Dou, Z., Suo, Y., Dabu, X. and Fu, H., 2018. Decreasing brown bear (Ursus arctos) habitat due to climate change in Central Asia and the Asian Highlands. Ecology and Evolution, 8(23), pp.11887-11899.

- Kafaei, S., Akmali, V. and Sharifi, M., 2020. Using the Ensemble Modeling Approach to Predict the Potential Distribution of the Muscat Mouse-Tailed Bat, Rhinopoma muscatellum (Chiroptera: Rhinopomatidae), in Iran. Iranian Journal of Science and Technology, Transactions A: Science, pp.1-12.

- Sharifi, M., Karami, P., Akmali, V., Afroosheh, M. and Vaissi, S., 2017. Modeling geographic distribution for the endangered yellow spotted mountain newt, Neurergus microspilotus (Amphibia: Salamandridae) in iran and Iraq. Herpetological Conservation and Biology, 12(2), pp.488-497.

In accordance with the opinion of the esteemed referee, the objectives of the article were modified in the introduction section.

Line 214: change ‘circuitscape’ to ‘Circuitscape’

[Authors’ response]: it was corrected. 

Line 214-218: This section would benefit from some more references to back up and provide further detail. 

[Authors’ response]: more references were added to support this section. 

Line 219: what is ‘The cost or Friction map’? would be added a citation! In addition, the authors sharply jumped into this paragraph without a logical transition from the later paragraph. It would be better rephrase these sections to follow the authors’ thoughts more easily. 

[Authors’ response]: More details about cost or friction map was added. Attempts were made to include the mentioned explanations in the text in an orderly and consistent manner in terms of meaning.

Line 221: What is a difference between the cost and resistance maps?

[Authors’ response]: They have no difference, and both point to the cost that the species must pay in the transmission path. 

Line 225: Add reference(s) to ‘or as the boundary of protected areas.’

[Authors’ response]: The revision was applied. This section was revised. Focal nodes in the software menu is a section that introduces habitat patches or connecting cores. In this study, protected areas were considered as focal nodes.

Line 225: What does mean by ‘focal nodes’?

 [Authors’ response]: The revision was applied. This part was revised due to incorrect translation. Focal nodes in the software menu is the part where the user introduces habitat spots or connection cores, which in this study, protected areas are considered as Focal nodes.

Lines 212-229: The use of ‘electronic circuit theory’ in the Methods needs more development in terms of recasting in a way that can be interpreted ecologically. Currently it reads very much as having applied a "black box" with foreign (to wildlife research at least) terms such as "current flow" and "resistance". Can these be recast as more accessible descriptions for the reader?

[Authors’ response]: The electrical circuit theory section was revised.

Line 231: I am really confused why the authors did employ a modeling technique (random forest) to see the contribution importance of the environmental variables in the species occurrence probability that was not included in an array of the applied modeling methods i.e. GLM, Maxent, SVM, and BP-ANN (lines 175-176) which were used to predict the species distribution.

[Authors’ response]: The habitat modeling software used in this study has several advantages, including having multiple modeling methods. In fact, this software offers many models used in the habitat modeling process. The ease of use and user-friendly graphical environment have led to easy use of this software. The software used in the model validation and threshold identification process works well, but there is no option to check the importance of the variables in its models. 

 

As shown in Figure 3, to determine the importance of different habitat variables, this software can only be implemented using a specific set of models.

Figure 3: Constraint of ModEco software in determining the importance of habitat variables

It was not possible to determine the importance of habitat variables in accordance with the models used in the modeling process using this software. Therefore, it was decided to use an alternative method that can appropriately determine the effect of habitat variables in each model. There are different methods in this field, but since random forest method is one of the widely-used methods (Cutler et al.,2007; Strobl et al.,2007) in identifying the importance of variables, it was used to investigate the importance of variables in each model.

- Cutler, D.R., Edwards Jr, T.C., Beard, K.H., Cutler, A., Hess, K.T., Gibson, J. and Lawler, J.J., 2007. Random forests for classification in ecology. Ecology, 88(11), pp.2783-2792.

- Strobl, C., Boulesteix, A.L., Zeileis, A. and Hothorn, T., 2007. Bias in random forest variable importance measures: Illustrations, sources and a solution. BMC bioinformatics, 8(1), p.25.

Lines 232-242: This section can be shortened more logically by adding more relevant refrence(s) to facilitate reader to follow the authors’ thoughts.

 [Authors’ response]: this section was revised as recommended. 

Line 241-242: This is extremely good to see the authors are using more applicable scientific program (R) here, by considering this point, I am really baffling why they did not use the potential of R program which is publically free available through more compressive species distribution analyzing packages to take into account the latest developments in species distribution modeling. I truly believe that this study will have meaningful contributions to the species ecology and conservation if the authors would polish their analyses in R!. 

[Authors’ response]: The words of the honourable referee are absolutely correct. However, there are reasons why authors tend to use MODECO software.

 1. Preparing pseudo-absence points and variety of existing methods:

Creating pseudo-absence points seamlessly is only possible in this software. This software provides the possibility to integrate different presence-only methods in order to achieve pseudo-absence points, which has not been included in various habitat modeling packages in the R environment.

2. Multiplicity of available 

In this software, more models are available for use than the packages in R software. In fact, almost all habitat modeling methods are included in this software.

3. Being fast and availability of different models:

In this software, the available models can be used freely and run very quickly.

4. Possibility to determine the type of forecast

In this software, it is possible for the user to determine the type of prediction as probable (probability of presence) or definite (zero and one). The following figure shows the available models for use.

Figure 4: models available in ModEco

Line 247 and Table 3: It is very interesting that I see the authors are using the significant metric (p-value) to address their findings. But it would be really useful to be added more descriptions in the methodology part of the MS how they did statistically compare the models performances using the evaluation metrics. This analysis would be more reproducible if the authors back their analysis given a reference(s). It is urgently needed to add more clarification on it since this is very important point. 

 [Authors’ response]: This part of the analysis is related to the norm hypothesis used to calculate the AUC, which was mentioned in the results section at the request of the Referee. 

Line 256: ‘the different RCP 2.6 scenarios’: It is not more clear. How many scenarios are available for RCP 2.6.? I assume that there is just one RCP 2.6! 

[Authors’ response]: This study considered three scenarios, including CCSM4, BCC-CSM1-1, and MRI-CGCM3 to predict future changes, each of which run under RCP2.6 and RCP8.5. So, the there are 3 RCP2.6 and 3 RCP8.5 in this study (Figure 3).

Line 258: ‘Under the RCP8.5 scenario, the habitat suitability declines in the northern parts…’. How do the authors come to this finding? It is worthy to be specific here by giving more information where and how many suitable patches will be lost under this scenario? AS the ms stands (lines 258-260), it is not scientifically informative to visually interpret the change trends of the species habitat suitability. 

[Authors’ response]: these are the presence probability maps with values ranging from 0 to 1 on a probability basis, so they cannot be analyzed quantitatively. This is why thresholds are applied to habitat suitability maps to quantify results. The complete and complementary results of this section are listed in Table 5.

Lines 265-266: How did the authors go straight to this point? Did they know where the suitable/unsuitable areas occur already to draw this interpretation?

[Authors’ response]: After identifying the cut-off value and applying it to the habitat suitability map, the map of suitable/unsuitable habitats were obtained. Then, the suitability values (1 and 0) of the habitat suitability map for the presence / pseudo-absence points were extracted. Accordingly, the metrics in Table 4 were easily calculated.

Table 4: What does mean by greater than for the thresholds of the model in the second column? Did the authors use these threshold levels to binaries the continuous suitability maps? If so, how? And also for what metrics (TSS, AUC, Kappa, …) was these levels calculated? It is really important to be awareness on uncertainties inherits in developing the models. So, the authors are strangely recommended to iterate each model several times (e.g. 30 times) to ensure that the results are expressed statistically meaningful in terms of the confidence intervals when it goes to consider the model performances using the elevation metrics. 

Answers to each questions:

1- What does mean by greater than for the thresholds of the model in the second column? Did the authors use these threshold levels to binaries the continuous suitability maps? If so, how?

[Authors’ response]: These threshold values are detected by SPSS software; and are usually identified as such. For example,> 0.26 shows the threshold at which 92% of the presence points are identified as presence points and 82% of the pseudo-non-presence points are identified as pseudo-non-presence points. This value was calculated based on the TSS threshold formula as follows (Allouche ET AL., 2006).

TSS= sensitivity + specificity – 1

Allouche, O., Tsoar, A. and Kadmon, R., 2006. Assessing the accuracy of species distribution models: prevalence, kappa and the true skill statistic (TSS). Journal of applied ecology, 43(6), pp.1223-1232.

This can be done using the raster calculator in the ArcMap software.

2- And also for what metrics (TSS, AUC, Kappa, …) was these levels calculated?

[Authors’ response]: Some of the mentioned metrics are threshold dependent and some are non-threshold dependent, which should be distinguished between the two groups of metrics. The AUC value is calculated using the ROC curve. The TSS statistic has two values, one related to validation and the other related to the threshold. Its numerical value in Table 3 can be interpreted with the explanations in the materials and methods section. Another value that can be calculated based on the TSS formula is the threshold, the exact value of which for each model run is listed in Table 4. After identifying the numerical value of the threshold limit, this value was applied on the suitability maps and the value of the resulting map (zero and one map obtained from the threshold) are obtained with the values of 0 and 1 of the pseudo-absence points and the values of the relevant metrics.

3- It is really important to be awareness on uncertainties inherits in developing the models. So, the authors are strangely recommended to iterate each model several times (e.g. 30 times) to ensure that the results are expressed statistically meaningful in terms of the confidence intervals when it goes to consider the model performances using the elevation metrics. 

[Authors’ response]: One of the reasons that Modeco software was chosen for modeling, as mentioned above, was its repeatability. In fact, the authors run these models for several times to get the best results. So, the valuable recommendation of the honourable Referee has already been applied when running the model.

Lines 277 and 278: What do the authors mean by the similarities? On base of what quantity metric(s) do they go for through the more similarities’ claim? 

[Authors’ response]: The mean was similarity between suitable and unsuitable habitats. This part was excluded from the results. 

-The terms ‘current flow’ (line 227) and ‘amps’ (line 299) seem a misnomer. Can you express in a more ecological manner that makes more sense to the reader? For example, the term ‘connectivity’ has an ecological meaning, but ‘current flow’ doesn’t at this stage.

 [Authors’ response]: In this section, an attempt was made to explain the results based on the ecological concepts and terms as recommended.

Line 323: What do bio19, 17, and 7 stand for? Here it is good to define these abbreviations. How do species response to these climatic variables. Creating response curves for them would add more value to the study.

[Authors’ response]: Table 2, which deals with habitat variables used, has been revised and the full name of the variables was added to the table. In order to prevent the increase of figures in the article, these figures are presented only in the Rebuttal letter not in the manuscript. The following figure shows the response curve of the 3 main influential variables.

Figure 4: Response curves of the 3 main variables affecting the species presence

Line 327: Capitalize ‘random’ 

[Authors’ response]: it was capitalized. 

-table 6: Remove unnecessary horizontal and vertical lines. What is ‘Peasant’? maybe you mean ‘present’. Uncertainty on these findings could be controlled if the authors performed multi replicates (e.g. 30 times) of their model as I mentioned before in developing species distribution modeling (please also note the comments on Table 4). 

 [Authors’ response]: the table was modified as recommended. One of the advantages of the software used in this study is replicability. The models mentioned in this study are the best models that are known to be appropriate in replication.

# General comments on figures: The figures are predominantly drawn in a way making it hard to interpret the main points what the authors would intend to present. Generally, what is highlighted in each figure exactly is not the main goal of the study but some main points that are needed to be more visible and clearer; unfortunately, they did not take into consideration for them. 

 [Authors’ response]: the figures of the paper have been revised as recommended.

---

## [Editor Report · Decision Letter 3]

3 Nov 2020

Accessing Habitat Suitability and Connectivity for the Westernmost Population of Asian Black Bear (Ursus thibetanus gedrosianus, Blanford, 1877) based on Climate Changes Scenarios in Iran

PONE-D-20-07352R3

Dear Dr. Morovati,

We’re pleased to inform you that your manuscript has been judged scientifically suitable for publication and will be formally accepted for publication once it meets all outstanding technical requirements.

Kind regards,

Lyi Mingyang, Ph.D.

Academic Editor

PLOS ONE

Additional Editor Comments (optional):

Well done!
---

## [Editor Report · Acceptance letter]

9 Nov 2020

PONE-D-20-07352R3 

**Accessing Habitat Suitability and Connectivity for the Westernmost Population of Asian Black Bear (*Ursus thibetanus gedrosianus*, Blanford, 1877) based on Climate Changes Scenarios in Iran******

****

**Dear Dr. Morovati:**

**I'm pleased to inform you that your manuscript has been deemed suitable for publication in PLOS ONE. Congratulations! Your manuscript is now with our production department. **

**If your institution or institutions have a press office, please let them know about your upcoming paper now to help maximize its impact. If they'll be preparing press materials, please inform our press team within the next 48 hours. Your manuscript will remain under strict press embargo until 2 pm Eastern Time on the date of publication. For more information please contact onepress@plos.org.**

**If we can help with anything else, please email us at plosone@plos.org. **

**Thank you for submitting your work to PLOS ONE and supporting open access. **

**Kind regards, **

**PLOS ONE Editorial Office Staff**

**on behalf of**

**Professor Lyi Mingyang **

**Academic Editor**

**PLOS ONE**